# The centromere landscapes of four karyotypically diverse *Papaver* species provide insights into chromosome evolution and speciation

## Graphical abstract

## Authors

Shenghan Gao (高胜寒), Yanyan Jia (贾彦彦), Hongtao Guo (郭弘涛), ..., Yimeng Zhang (张一蒙), Xiaofei Yang (杨晓飞), Kai Ye (叶凯)

## Correspondence

xfyang@xjtu.edu.cn (X.Y.), kaiye@xjtu.edu.cn (K.Y.)

## In brief

Gao et al. assembled four near-complete *Papaver* genomes and characterized the diversity of their centromeric landscapes. Their results advance *Papaver* as a new model for studying centromere evolution and, in particular, highlight the extensive centromere-mediated chromosomal rearrangements.

## Highlights

- Near telomere-to-telomere genome assemblies of four *Papaver* species

- Extensive and complex centromere-mediated chromosome rearrangements

- Centromere satellite homogenization following hybridization

- *Papaver* as a novel model for studying centromere evolution

Gao et al., 2024, Cell Genomics *4*, 100626
August 14, 2024 © 2024 The Author(s). Published by Elsevier Inc.

CellPress

# The centromere landscapes of four karyotypically diverse *Papaver* species provide insights into chromosome evolution and speciation

Shenghan Gao (高胜寒),[1,2,3,8] Yanyan Jia (贾彦彦),[1,8] Hongtao Guo (郭弘涛),[2] Tun Xu (徐曦),[1,3] Bo Wang (王博),[1] Stephen J. Bush,[1] Shijie Wan (万世杰),[2] Yimeng Zhang (张一蒙),[2] Xiaofei Yang (杨晓飞),[2,3,*] and Kai Ye (叶凯)[1,3,4,5,6,7,9,*]

[1]School of Automation Science and Engineering, Faculty of Electronic and Information Engineering, Xi'an Jiaotong University, Xi'an, Shaanxi 710049, China
[2]School of Computer Science and Technology, Faculty of Electronic and Information Engineering, Xi'an Jiaotong University, Xi'an, Shaanxi 710049, China
[3]MOE Key Lab for Intelligent Networks & Networks Security, Faculty of Electronic and Information Engineering, Xi'an Jiaotong University, Xi'an, Shaanxi 710049, China
[4]Center for Mathematical Medical, The First Affiliated Hospital, Xi'an Jiaotong University, Xi'an, Shaanxi 710061, China
[5]Genome Institute, The First Affiliated Hospital, Xi'an Jiaotong University, Xi'an, Shaanxi 710061, China
[6]School of Life Science and Technology, Xi'an Jiaotong University, Xi'an, Shaanxi 710049, China
[7]Faculty of Science, Leiden University, Leiden 2311EZ, the Netherlands
[8]These authors contributed equally
[9]Lead contact
*Correspondence: xfyang@xjtu.edu.cn (X.Y.), kaiye@xjtu.edu.cn (K.Y.)

## SUMMARY

Understanding the roles played by centromeres in chromosome evolution and speciation is complicated by the fact that centromeres comprise large arrays of tandemly repeated satellite DNA, which hinders high-quality assembly. Here, we used long-read sequencing to generate nearly complete genome assemblies for four karyotypically diverse *Papaver* species, *P. setigerum* (2*n* = 44), *P. somniferum* (2*n* = 22), *P. rhoeas* (2*n* = 14), and *P. bracteatum* (2*n* = 14), collectively representing 45 gapless centromeres. We identified four centromere satellite (cenSat) families and experimentally validated two representatives. For the two allopolyploid genomes (*P. somniferum* and *P. setigerum*), we characterized the subgenomic distribution of each satellite and identified a "homogenizing" phase of centromere evolution in the aftermath of hybridization. An interspecies comparison of the peri-centromeric regions further revealed extensive centromere-mediated chromosome rearrangements. Taking these results together, we propose a model for studying cenSat competition after hybridization and shed further light on the complex role of the centromere in speciation.

## INTRODUCTION

Centromeres are fundamental chromosomal elements with an essential role in chromosome segregation at cell division[1–3] and are considered sources of genomic instability[4,5] that mediate speciation.[6,7] Centromeres of many species comprise large arrays of tandemly repeated satellite DNA often dominated by one repeat, such as alpha satellite DNA in human[8–10] and *AthCEN178* (formerly *CEN180*) in *Arabidopsis thaliana*.[11,12] This repetitive nature of the centromere complicates their assembly and, thereby, investigation into their evolutionary history.[13,14] Nevertheless, recent advances in assembly methods and long-read DNA sequencing technologies, including PacBio high-fidelity (HiFi) and Oxford Nanopore Technology (ONT), have facilitated the complete assembly of complex centromeres, producing both telomere-to-telomere (T2T) genome assemblies for human and *Arabidopsis thaliana*,[10,11,15] the species in which centromeres have been most widely studied to date. The near-universal dominance by one type of centromere satellite (cenSat) is of note given the centromere drive hypothesis, which proposes that during asymmetric female meiosis (when an entire set of chromosomes is discarded), centromere variants can "cheat" the process, acting as selfish genetic elements and competing for inclusion in the daughter cell.[1,2,16] It follows that for species whose centromeres comprise multiple cenSat families, rather than just one, "drive" in meiosis may have a more elaborate presentation. Consistent with this, a growing body of experimental evidence suggests a complex role for the centromere in shaping the genome, for instance through centromere-mediated translocations "shuffling" the karyotype,[17] among other gross chromosomal rearrangements.[18] However, this interplay between cenSats and chromosome rearrangements remains poorly understood due to the lack of complete centromere assemblies for closely related species under complex evolutionary scenarios.

**Cell Genomics**
**Article**

**Table 1. Genome assembly and annotation statistics**

|  | *P. setigerum* | *P. somniferum* | *P. rhoeas* | *P. bracteatum* |
|---|---|---|---|---|
| Total assembly size (Mb) | 4,691.55 | 2,786.51 | 2,291.05 | 2,472.23 |
| Contig N50 (Mb) | 129.23 | 214.79 | 29.07 | 331.46 |
| Largest contigs (Mb) | 256.96 | 344.13 | 202.12 | 365.70 |
| Scaffold N50 (Mb) | 217.39 | 261.98 | 295.71 | 351.02 |
| Sequences anchored to chromosome (%) | 98.7 | 98.3 | 94.7 | 99.0 |
| No. of gaps | 48 | 22 | 138 | 18 |
| Genome completeness (BUSCO) (%) | 98.7 | 98.2 | 94.1 | 97.9 |
| No. of protein-coding genes | 126,422 | 64,087 | 42,133 | 40,371 |
| Repeat density (%) | 73.60 | 74.42 | 73.51 | 73.47 |

See also Figures S2–S5 and Tables S2, S3, and S4.

To address this issue, and thereby provide further insight into the forces shaping chromosome evolution, we investigated species of the *Papaver* genus, as they have experienced multiple rounds of allopolyploidization[19,20] and so have a variety of karyotypes.[21] By using HiFi, ONT, and high-throughput chromosome conformation capture (Hi-C) reads, we generated near-T2T genome assembles of four *Papaver* species (*P. setigerum* [2*n* = 44], *P. somniferum* [2*n* = 22], *P. rhoeas* [2*n* = 14], and *P. bracteatum* [2*n* = 14]); characterized the genomic landscape of their cenSats; and—by comparing syntenic chromosome pairs—identified extensive centromere-mediated chromosome rearrangements (CMCRs). In conjunction with both inter- and intra-species comparisons of cenSat diversity, we advance *Papaver* as a new model for studying centromere evolution and provide new insights into the complex role of centromeres in speciation.

## RESULTS

### Genome assembly and annotation

We sequenced DNA from the leaves of four *Papaver* species, *P. setigerum* (2*n* = 44), *P. somniferum* (2*n* = 22), *P. rhoeas* (2*n* = 14), and *P. bracteatum* (2*n* = 14) (Figure S1), to an average depth of 47× (ranging from 35× to 58×) of PacBio HiFi long reads and 134× of Hi-C short reads (Table S1). Using a custom genome assembly pipeline (Figure S2), we assembled four near-T2T genomes with total sizes of 4.7 Gb for *P. setigerum*, 2.8 Gb for *P. somniferum*, 2.3 Gb for *P. rhoeas*, and 2.5 Gb for *P. bracteatum*, representing, on average, 97.6% of each genome's estimated size[19,22] (Tables 1 and S2; Figure S3). The mean contig N50 for the four assemblies was 176.1 Mb (Table S2). The contig N50s are 221-, 123-, 5-, and 2-fold those of previous assemblies for *P. bracteatum*, *P. somniferum*, *P. rhoeas*, and *P. setigerum*, respectively[19,23] (Figure S4; Table S2). We anchored an average of 97.7% of the assembled sequence to chromosomes using Hi-C scaffolding and obtained a mean scaffold N50 of 281.5 Mb (Tables 1 and S2; Figure S5). In total, 73 out of 94 (77.7%) telomeres were assembled with the motif 5′-CCCTGAA-3′ (Figures S6–S8). Relative to existing *Papaver* assemblies, the number of gaps in each genome

decreased from thousands or hundreds to tens; for example, in *P. somniferum*, we reduced the number of gaps from 6,421 to 22,[19] and for three chromosomes (chr4, 8, and 10), we achieved gapless T2T assembly (Tables 1 and S2; Figure S6A).

We aligned both short and long reads to each assembly at an average mapping rate of 99.6% (Table S3). The average base accuracy of the four genomes was estimated as 99.998% (quality value score, 48.2) (Table S2). The completeness of the four genomes was evaluated using Benchmarking Universal Single-Copy Orthologs,[24] with an average completeness score of 97.2% (Table S2). Together, these results indicate the high quality of the four *Papaver* genome assemblies.

Approximately three-quarters of each of the four *Papaver* genomes comprised repetitive elements, with long terminal repeat (LTR) retrotransposons the predominant repeat family, comprising, on average, 54% of the genome (Tables 1 and S4). Each of the four genomes was broadly consistent in its composition of LTR retrotransposons, with Gypsy elements accounting for 22.8%–27.5% of the genome and Copia elements 15.6%–25.1% (Table S4). However, we observed that *P. bracteatum* was at the extremes of each range, having a genome comprising 15.6% Copia and 27.5% Gypsy elements, suggesting a relative contraction of the former and expansion of the latter (Table S4). In addition, we found that, on average, only 4.5%, 3.7%, and 0.04% of the genome could be annotated as DNA transposons, long interspersed nuclear elements (LINEs), and short interspersed nuclear elements (SINEs), respectively (Table S4). We next annotated an average of 21,090 non-coding RNAs in the four *Papaver* species and predicted 126,422, 64,087, 42,133, and 40,371 protein-coding genes in *P. setigerum*, *P. somniferum*, *P. rhoeas*, and *P. bracteatum*, respectively, after integrating evidence from protein homology, RNA sequencing (RNA-seq), and *ab initio* prediction (Tables 1 and S4). Each gene was supported by either homology or transcript-level evidence, with 64.2%–66.0% functionally annotated by InterProScan[25] (Table S4).

### Centromere prediction and validation

We annotated the tandem repeats (TRs), a defining characteristic of centromeric DNA, using TR Finder[26] and found that, as expected, TRs were disproportionately located at specific

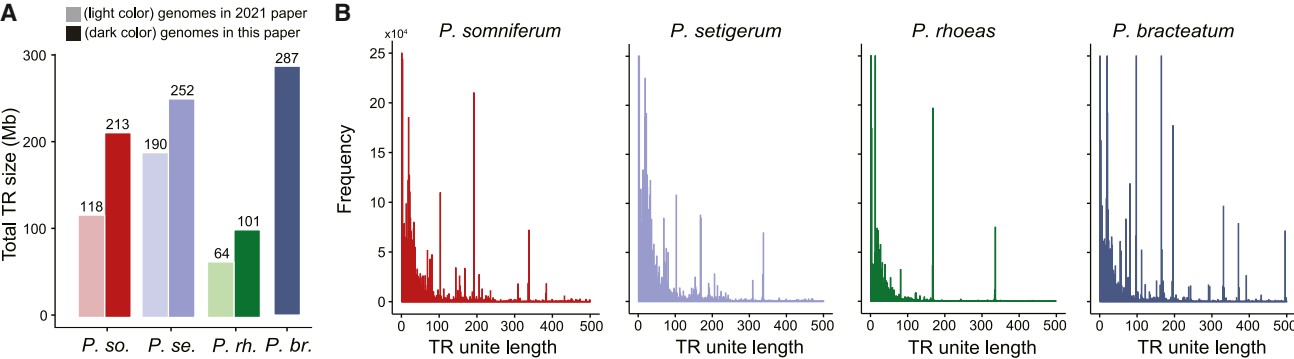

**Figure 1. TR annotations in four *Papaver* species**
(A) The total annotated tandem repeat (TR) size in four species. We compared the total TR sizes between the genomes in this work and that published in 2021.[19] Light colors are related to the 2021 paper, and dark colors are related to this paper.
(B) The TR unit length distribution in four species. *P. so.*, *P. somniferum*; *P. se.*, *P. setigerum*; *P. rh.*, *P. rhoeas*; *P. br.*, *P. bracteatum*.
See also Figures S6–S8.

genomic regions (Figures S6–S8), ostensibly the centromeres. The summed length of TRs was significantly increased relative to previous assemblies,[19] totaling 100.7, 213.2, 252.5, and 287.5 Mb in *P. rhoeas*, *P. somniferum*, *P. setigerum*, and *P. bracteatum*, respectively (Figure 1A). The per-species TR unit length distributions varied in their number of peaks (Figure 1B), suggesting a diverse landscape of centromeric sequences, although *P. somniferum* and *P. setigerum* had notably similar distributions, consistent with their close phylogenetic relationship (Figure S9; Table S5).

To identify cenSats from the annotated TRs, we applied a community detection method[27,28] to the TR similarity network, producing a satellite DNA library with a set of 124 satellites (irrespective of genomic location), which we ranked on the basis of AT content (as the centromeric sequence is often characteristically AT rich[2]) (Figure S10; Table S6). After excluding satellites with a genomic size < 1 Mb or AT content < 60%, we produced a library of 26 candidate cenSats, of which 7 were found in *P. bracteatum*, 8 in *P. setigerum*, and 10 in *P. somniferum* but only one in *P. rhoeas*, Prh168S1 (a 168 bp repeat unit with total genomic size 31.2 Mb) (Table S6). We validated Prh168S1 as a *bona fide* cenSat in *P. rhoeas* using fluorescence *in situ* hybridization (FISH), observing a strong signal at each centromere in metaphase (Figures 2A, S11A, and S11B). Furthermore, we performed chromatin immunoprecipitation followed by sequencing (ChIP-seq) with the *P. rhoeas* centromere-specific histone H3 (PrhCENH3) antibody (centromeric histone H3, encoded by *P. rhoeas* gene *Prh03G45160.1*) (Figure S12) and aligned the data to the assembled genome. We observed, on average, 11-fold ChIP/input enrichment within the identified Prh168S1 arrays compared with the other genomic regions, supporting our interpretation of Prh168S1 as a cenSat in *P. rhoeas* (Figure S13; Table S7).

Accordingly, we filtered the remaining 25 candidate cenSats based on their pairwise syntenic relationships with the peri-centromeric regions of *P. rhoeas* (Figures 2B, S14, and S15; Tables S8 and S9). We found that there was one syntenic cenSat in *P. somniferum* (Pso338S1, total size of 23.9 Mb, detected on all

11 chromosomes), two in *P. bracteatum* (Pbr238S1, total size of 7.3 Mb, detected on four of seven chromosomes; and Pbr169S4, total size of 1.9 Mb, detected only on chr1), and four in *P. setigerum* (Pse338S1, Pse168S7, Pse169S11, and Pse168S13) (Figures 2B, S14, and S15; Tables S8, S9, and S10). As with *P. rhoeas*, we validated Pso338S1 as a *bona fide* cenSat in *P. somniferum* using FISH, again observing a strong signal for each centromere in metaphase (Figures 2C, S11C, and S11D). To complement this finding, we also performed ChIP-seq with the PsoCENH3 antibody (centromeric histone H3, encoded by *P. somniferum* gene *Pso04G02820.1*) (Figure S16) and aligned the data to the assembled genome, finding significantly higher read coverage at the Pso338S1 locus in each of the 11 chromosomes and supporting our interpretation of Pso338S1 as a prevalent cenSat in *P. somniferum* (Figures 2D and S17). Specifically, we observed, on average, 4-fold ChIP/ input enrichment within the identified Pso338S1 array regions, compared with the other genomic regions (Table S10). A number of centromeres, including chr9, 10, and 11, also showed multiple discrete PsoCENH3 peaks (Figure 2D), suggesting that they either contained multiple discrete centromere regions or, alternatively, that individual centromeres from different cells were merged in our population-scale analysis. We analyzed the repetitive elements in the centromeric and peri-centromeric regions, observing a higher prevalence of LTR-Gypsy and LINE compared to LTR-Copia within the PsoCENH3 occupied regions. Moreover, the cenSat arrays are characterized by a relative scarcity of genes (Figure 2D). We further validated the centromere assemblies with long-read coverage and VerityMap[29] (Figures S18 and S19), which collectively indicated their quality and completeness. We next examined the cenSats of *P. setigerum* and found that Pse338S1 and Pse168S7 shared 99% and 94% identity with the experimentally validated cenSats Pso338S1 and Prh168S1, respectively (Table S11).

## Genetic landscape of cenSat arrays
To obtain a global view of cenSats across the four *Papaver* species, we constructed a cross-species satellite similarity

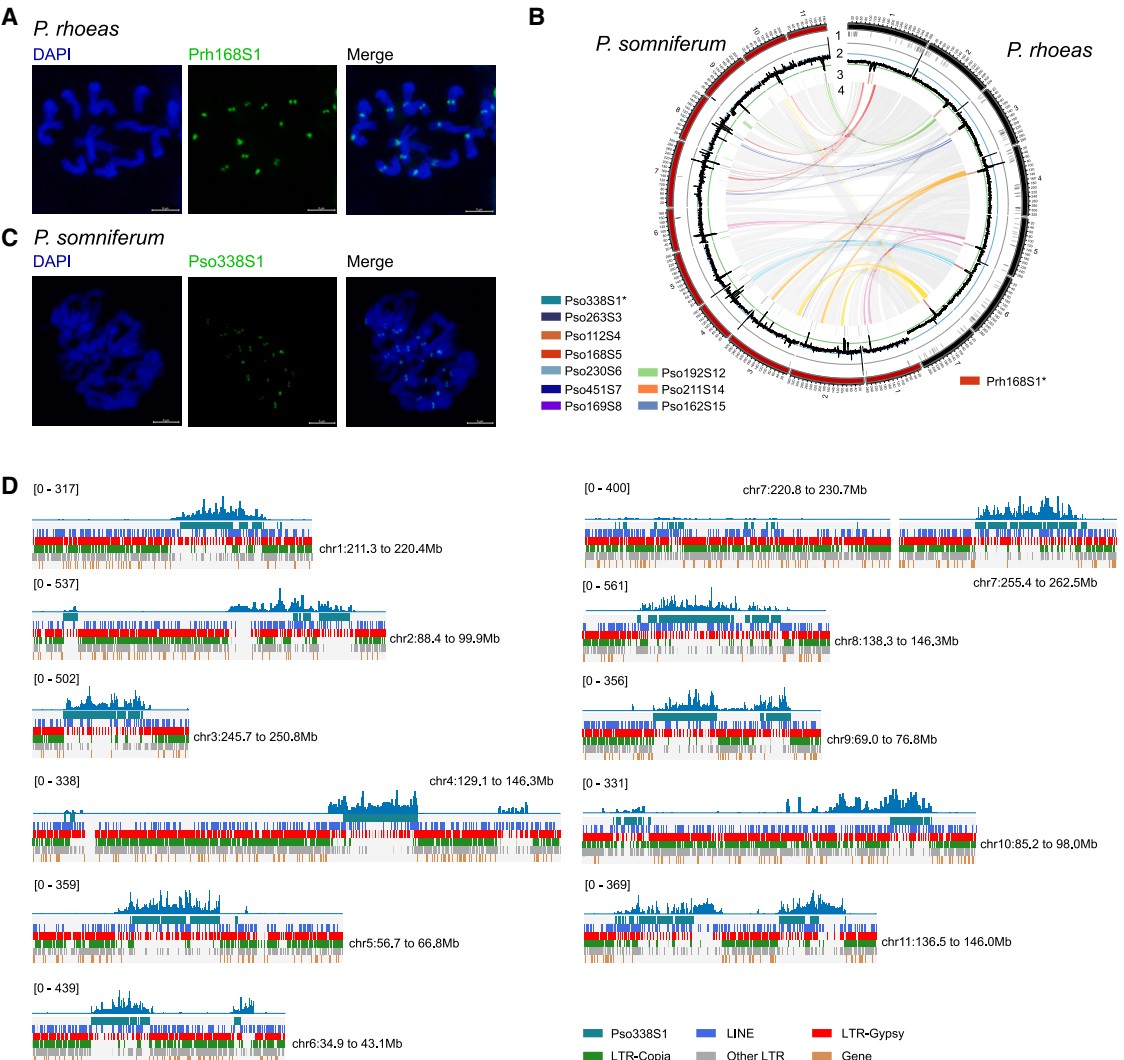

**Figure 2. Prediction, validation, and characterization of complete centromeres**

(A) Fluorescence *in situ* hybridization (FISH) of Prh168S1 on metaphase chromosomes in *P. rhoeas*. Metaphase chromosomes stained with 4′,6-diamidino-2-phenylindole (DAPI) (blue, left) and Prh168S1 FISH probes (green, middle).

(B) Centromere satellite (cenSat) prediction based on peri-centromeric syntenic relations. The tracks numbered 1, 2, 3, and 4 indicate assembly gaps, TR density, satellites annotations, and syntenic relationships, respectively. The peri-centromeric syntenic relationships are highlighted with different colors. Satellites with "*" represent the predicted cenSats.

(C) FISH of Pso338S1 on metaphase chromosomes in *P. somniferum*. Metaphase chromosomes stained with DAPI (blue, left) and Pso338S1 FISH probes (green, middle).

(D) Sequence coverage of PsoCENH3 ChIP-seq data on each inferred (peri-)centromere region in *P. somniferum*. Tracks from bottom to top indicate the location of genes, transposable elements (other LTRs, LTR-Copias, LTR-Gypsys, and LINEs), and Pso338S1 satellite arrays, respectively.

See also Figures S10–S19 and Tables S6, S7, S9, and S10.

network and detected one isolated node and five communities, represented by αPCEN169, PCEN238, αPCEN168, PCEN338, βPCEN169, and βPCEN168 (Figure 3A; Table S11). The genomic maps of the detected cenSats in *P. bracteatum*, *P. rhoeas*, *P. setigerum*, and *P. somniferum* were shown in Figures 3B, 3C, 3D, and 3E, respectively. We found that with the exception of βPCEN168, every cenSat was found in multiple species but at notably different levels in each. For instance, we detected 188,050, 63,070, and 30,048 copies of αPCEN168 in *P. rhoeas*,

*P. setigerum*, and *P. somniferum*, respectively (Figure 3A; Table S12). Four communities (represented by PCEN338, αPCEN168, PCEN238, and αPCEN169 and hereafter defined as representative cenSats) were substantively detected (copy number > 1,000) in multiple chromosomes or species (Figures 3A–3E; Table S12). Three of these communities, represented by PCEN338, αPCEN168, and αPCEN169, were found in *P. setigerum* with 53,592, 63,070, and 31,166 copies, respectively (Table S12), and correspond to three chromosome groups

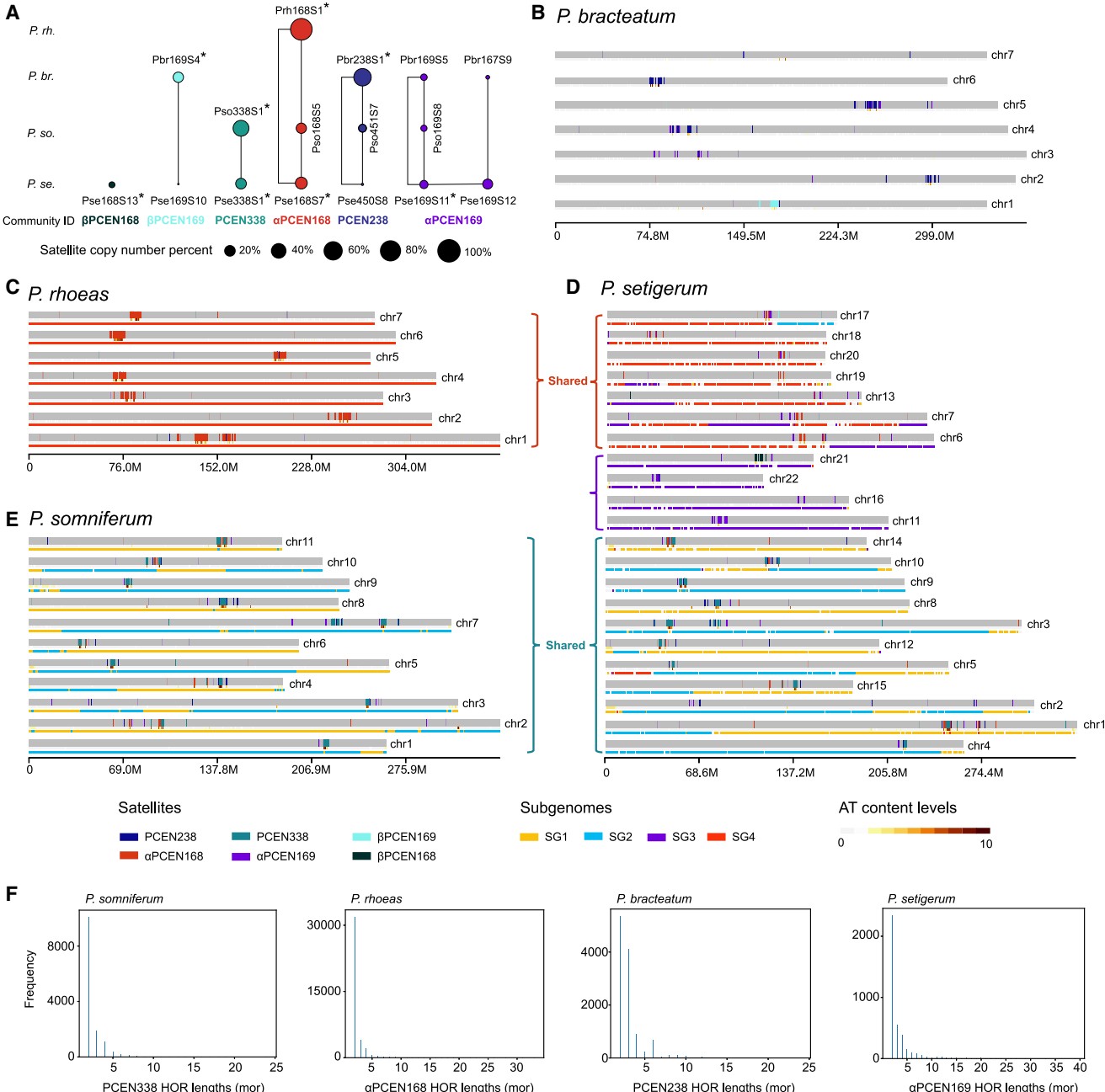

**Figure 3. Genomic landscape of *Papaver* cenSat arrays**

(A) CenSat communities detected using a cross-species satellite similarity network. Satellites with "*" represent predicted cenSats based on peri-centromere synteny. Satellites in four communities (represented by PCEN338 [teal], αPCEN168 [red], PCEN238 [dark blue], and αPCEN169 [purple]) amplified in more than one species and multiple chromosomes are defined as representative cenSats.

(B–E) The genomic landscape of cenSats in *P. bracteatum* (B), *P. rhoeas* (C), *P. setigerum* (D), and *P. somniferum* (E). The most abundant cenSats in *P. bracteatum*, *P. rhoeas*, and *P. somniferum* are PCEN238 (dark blue vertical bars), αPCEN168, and PCEN338, respectively. In the allopolyploid *P. setigerum*, multiple cenSats (PCEN338, αPCEN168, and αPCEN169) can be observed at varying densities. The AT content is divided into 10 levels and indicated by shading, with more AT in darker shades. Subgenome phasing results are indicated as color bars below each chromosome. We obtained four subgenomes (SG1, SG2, SG3, and SG4) from *P. somniferum* and *P. setigerum* and found that SG1 and SG2 were shared between the two. As the cenSat Prh168S1 was shared between *P. rhoeas* and SG4, we infer that *P. rhoeas* and *P. setigerum* shared SG4.

(F) Higher-order repeat (HOR) unit length distribution of PCEN338, αPCEN168, PCEN238, and αPCEN169 arrays in *P. somniferum*, *P. rhoeas*, *P. bracteatum*, and *P. setigerum*, respectively.

See also Figures S20–S22 and Tables 11, S12, and S13.

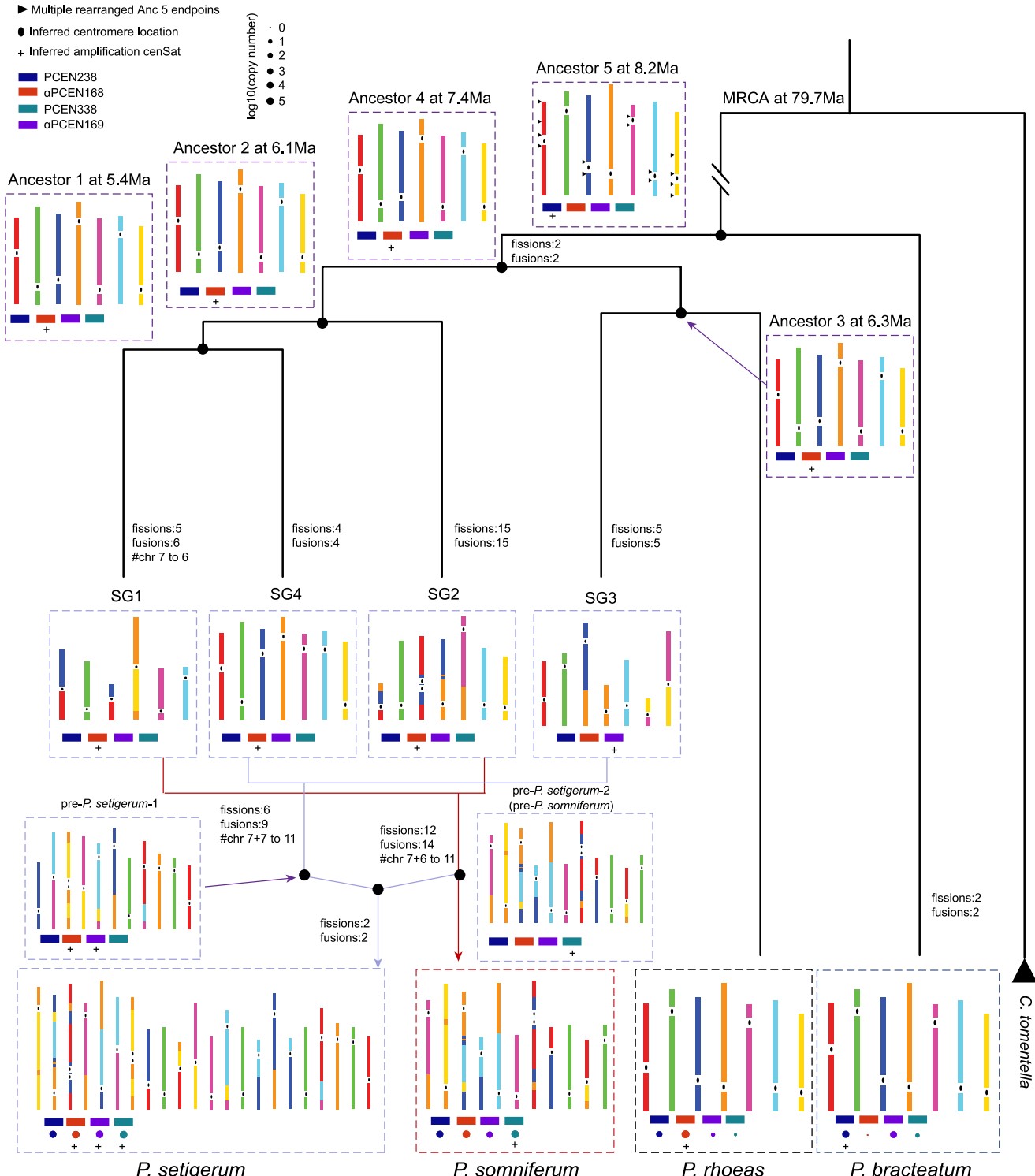

**Figure 4. Subgenome-aware phylogenetic tree charting the evolutionary history of *Papaver* cenSats and karyotypes**

The evolutionary history of four representative cenSats (PCEN338, αPCEN168, PCEN238, and αPCEN169, each representative of a community in Figure 3A) was inferred based on their existence in four modern genomes. "+" sign indicates the inferred amplified satellites, with the relative size of each circle indicating the total copy number of each satellite in each of the four modern *Papaver* species. Ancestral karyotypes were reconstructed using IAGS,[32] with centromere locations inferred from syntenic relationships between the ancestral and extant genomes. Each of the seven chromosomes of ancestor 5, the most recent common ancestor (MRCA) of the four species in this study (top center), was colored separately. The color composition of all chromosomes descending from ancestor 5

*(legend continued on next page)*

(Figure 3D). We annotated the higher-order repeat (HOR) structures of the four representative cenSat arrays in their corresponding species, e.g., PCEN338 in *P. somniferum*, αPCEN168 in *P. rhoeas*, αPCEN169 in *P. setigerum*, and PCEN238 in *P. bracteatum*, by applying HiCAT.[30] We identified HORs in each species and found a negative exponential distribution of HOR lengths, consistent with previous observations in *Arabidopsis thaliana*[11,12] (Figure 3F) and suggesting that cenSats expand in a similar manner in both *Arabidopsis* and *Papaver*. The isolated node βPCEN168 was only detected on *P. setigerum* chr21 (with 15,941 copies), while βPCEN169 expanded primarily on *P. bracteatum* chr1 with 11,370 copies (Figures 3B and 3D; Table S12), suggesting in both cases that they may be neo-cenSats (i.e., "neo-arrays"). Moreover, some chromosomes appeared "seeded" with trace numbers of cenSats, for instance *P. setigerum* chr2 (826 copies in total) and *P. bracteatum* chr7 (237 copies in total) (Figures 3B and 3D; Table S12).

We next compared syntenic chromosome pairs between different species and observed that chromosomal rearrangements in the vicinity of centromeres were involved in the expansion and contraction of satellite arrays (Figures S20–S22). For instance, a deletion-inversion event contributed to the loss of the PCEN338 array from *P. setigerum* chr2 (Figure S21), and an inversion event in *P. somniferum* chr11 rearranged the satellite array compared to its syntenic counterpart, *P. setigerum* chr14 (Figure S22). Repeat elements, including LTRs and LINEs, were observed around these variations (Table S13). Furthermore, we detected different levels of chromosomal shuffling around centromeres associated with their array patterns (Figures S20A–S20C). Specifically, syntenic chromosome pairs between *P. rhoeas* and *P. bracteatum* with conserved centromeric loci show overall syntenic conservation with small-scale (about 15.8–43.6 Mb in size) chromosomal shuffling in the vicinity of the cenSats (Figure S20A), while those chromosome pairs with seeded arrays showed massive (about 112.1–125.0 Mb in size) shuffling events (Figure S20B) and evident transposable element insertion (Figures S20D and S20E).

## Centromere and subgenome-aware ancestral state reconstruction

The diversity of the *Papaver* cenSats and their species-specific non-random distribution prompted us to investigate how they contribute to the evolution of karyotypes. To that end, we first phased the subgenomes of the two allopolyploid species, *P. setigerum* and *P. somniferum*, by SubPhaser,[31] obtaining four subgenomes, SG1, SG2, SG3, and SG4, of which SG1 and SG2 were held in common (Figures 3D and 3E). Subgenome-aware phylogenomic analysis indicated that hybridization between both SG1 and SG2 and SG3 and SG4 formed pre-*P. setigerum*-1 and pre-*P. setigerum*-2 (also pre-*P. somniferum*), respectively, consistent with the reticulate allopolyploid origin of *P. setigerum*[20] (Figure 4). Interestingly, we observed different compositions of cenSats in each subgenome.

For instance, SG1 and SG2 share PCEN338 (71,164 copies in *P. somniferum* and 53,587 copies in *P. setigerum*), SG4 and *P. rhoeas* share αPCEN168 (35,615 copies in SG4 and 188,050 copies in *P. rhoeas*), and SG3 amplifies αPCEN169 (24,625 copies) (Figures 3D and 3E; Table S12).

We next reconstructed the probable trajectories of cenSat expansion and replacement (Figure 4). Due to its presence in all four *Papaver* species, PCEN238 was likely the oldest amplified cenSat, i.e., before 8.2 million years ago (mya). Then, the αPCEN168 array replaced the PCEN238 array after the divergence of ancestor 4 and *P. bracteatum* at about 7.4 mya. The αPCEN169 array was likely amplified in an SG3-linage-specific manner, whereas the PCEN338 array was the most recent of the four to expand, after the hybridization of SG1 and SG2 (Figure 4). This inferred sequential order of cenSat expansions is supported by the observation of a symmetric composition of cenSats in *P. setigerum* chr1, for example of PCEN338 recurrently flanked with αPCEN168 and PCEN238 on both sides, following a "layered expansion" model of cenSat evolution (whereby newly inserted repeats expand within, and displace, the existing "layer" of repeats)[10,33] (Figure 3D).

To investigate whether centromeres contribute to the formation of karyotypes, we reconstructed five ancestral karyotypes using the IAGS framework[32] and inferred the numbers of chromosomal fissions and fusions between the adjacent evolutionary states (Figures 4 and S23; Tables 14 and S15). The total number of chromosomes for all five inferred ancestors and three subgenomes was seven, consistent with a previous study based on karyomorphology.[21] Three branches contained unbalanced numbers of fissions and fusions, resulting in changes of chromosome number; for example, six fissions and nine fusions after the hybridization of SG4 and SG3 reduced the chromosome number from 14 to 11 in pre-*P. setigerum*-2 (pre-*P. somniferum*). Four subgenomes and *P. rhoeas* probably shared the same ancestor (ancestor 4), and as we did not detect any fissions or fusions between ancestor 4 and *P. rhoeas*, this suggests that the *P. rhoeas* genome most closely resembles ancestor 4 (Figure 4). We summarized the multiple rearrangement endpoints of the four species in this study compared to their most recent common ancestor (MRCA) (ancestor 5) and found that most of the breakpoints (71.4%, 10 out of 14) were located around the centromeres inferred in ancestor 5, indicating an important role of centromeres in chromosome rearrangement (Figure 4).

## CMCR

To further explore the relationship between centromeres and chromosomal rearrangement, we analyzed the conservation of synteny at centromeric loci between the four subgenome-phased chromosomes of *P. setigerum* and their MRCA (ancestor 4, represented by *P. rhoeas*). We observed six types of chromosomal rearrangement, affecting 15 chromosomes, nine of which involved a centromeric sequence (Figures 5A and S24). These

---

indicate the inheritance of rearrangements relative to it. The total number of fission and fusion events, along with changes to total chromosome number, are labeled on the tree. The locations of the multiple rearrangement endpoints observed in each of the four extant species relative to ancestor 5 are indicated by black triangles on the chromosomes of ancestor 5.

See also Figure S23 and Tables 14 and S15.

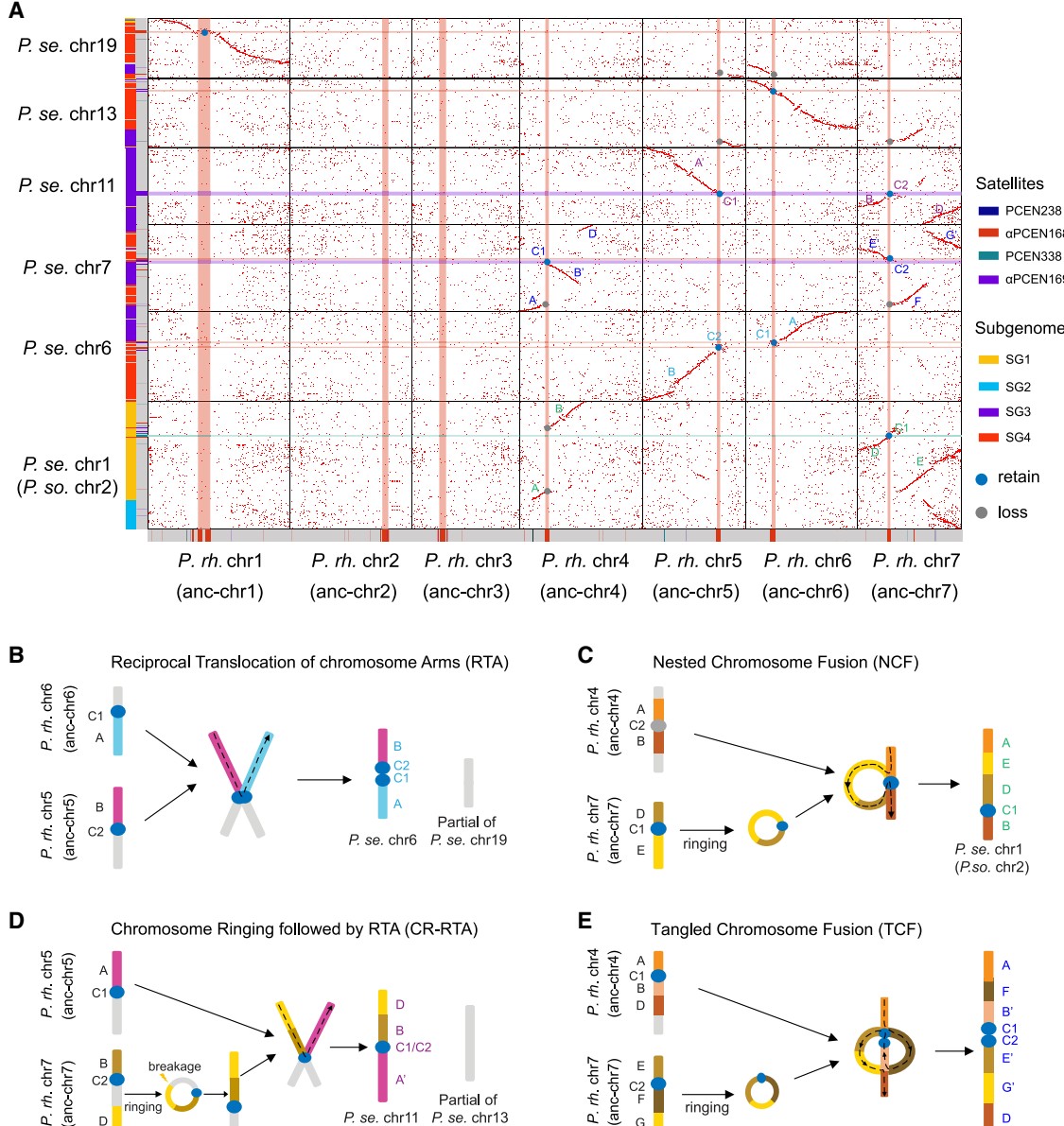

**Figure 5. Four types of centromere-mediated chromosome rearrangement**

(A) Syntenic dot plot between *P. setigerum* and the MRCA of its four subgenomes (labeled as ancestor 4 in Figure 4, although for ease of representation, its chromosomes are numbered according to those in *P. rhoeas*). Black grid lines separate the chromosomes in the two species. CenSat locations are shown on each chromosome and highlighted as both horizontal and vertical bars through the dot plot. Subgenome phasing results for *P. setigerum* are shown on the y axis as color bars to the left of each chromosome and follow the same color scheme as Figure 3. The retention or loss of syntenic cenSats was marked by blue or gray dots, respectively. The whole dot plot is in Figure S24.

(B–E) The formation of *P. setigerum* chr6 (B), chr1 (C), chr11 (D), and chr7 (E) following reciprocal translocation of chromosome arms (RTA) (B), nested chromosome fusion (NCF) (C), chromosome ringing followed by RTA (CR-RTA) (D), and tangled chromosome fusion (TCF) (E), respectively. Chromosome segment IDs correspond to those in (A). *P. se.*, *P. setigerum*; *P. so.*, *P. somniferum*; *P. rh.*, *P. rhoeas*.

See also Figures S24–S29.

observations support a possible relationship between the centromere dynamics of the subgenomes and structural rearrangements within them, one more complex than (for instance) that observed in *Arabidopsis suecica*, in which there was little evidence of "genomic shock" following its polyploidization, with no

subgenome dominance in expression, seemingly isolated cenSats, and no apparent genomic reorganization.[34]

Among the *Papaver* chromosomal rearrangements, we found that the formation of *P. setigerum* chr5, 6, and 17 were the products of a reciprocal translocation of chromosome arms (RTA)

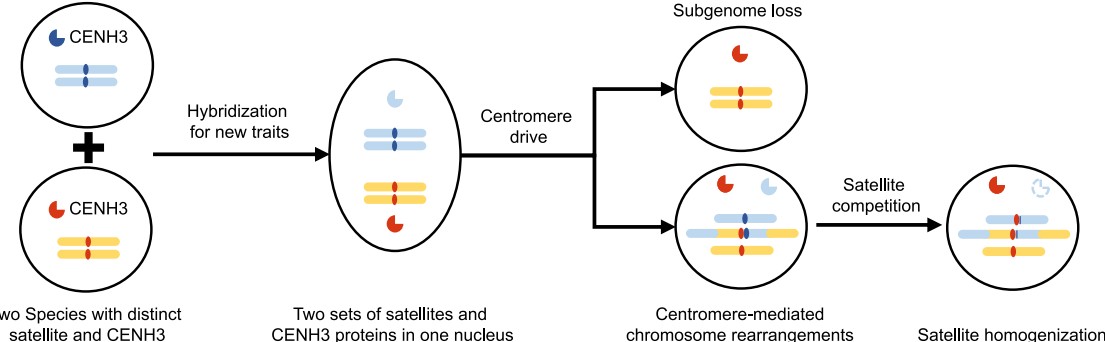

**Figure 6. A model of centromere evolution following hybridization**

Following the hybridization of two species with different cenSats and CENH3 proteins (blue and red), these coexist in the same nucleus. As a consequence of centromere drive and differential expression of the two *CENH3* genes, one set may preferentially accumulate over time, leading to either the entire or partial loss of the other (here we assume that the red *CENH3* gene is more highly expressed than that the blue one). Alternatively, chromosomal rearrangement in the vicinity of the centromere could bring two rival satellites together, with competition between them promoting fixation (homogenization) of the winner (red). The defeated cenSat and/or its corresponding *CENH3* may be degraded, downregulated, or lost from the genome entirely.

See also Figure S30.

(Figure S24). More specifically, the long arms of ancestor chr5 (anc-chr5) and anc-chr6 were joined at the centromeres, forming *P. setigerum* chr6 (Pse-chr6), with the two short arms joining with a short fragment of the αPCEN168 cenSat (125 copies) to form a part of Pse-chr19 (Figures 5B and S25; Table S12). In addition to linear chromosome joining, we also observed three ring-chromosome-related types of rearrangement, including nested chromosome fusion (NCF),[35] chromosome ringing preceding RTA (CR-RTA), and tangled chromosome fusion (TCF). Specifically, the long arm of Pse-chr1 formed through NCF of anc-chr4 and ringed anc-chr7, resulting in a truncation at the anc-chr4 centromere and the formation of a residual αPCEN168 cenSat (3,480 copies) (Figures 5C and S26; Table S12). We also observed that a complex NCF, containing (alongside a ''classical'' NCF) both sequential rearrangement events from homologous recombination and local translocations, produced Pse-chr3 and *P. somniferum* chr7 (Pso-chr7) (Figure S27). CR-RTA lead to the formation of Pse-chr11 when anc-chr7 formed a ring structure, broke, and then merged with anc-chr5 at the centromere locus in a manner resembling RTA (Figures 5D and S28). We propose TCF as a novel form of complex chromosomal rearrangement based on the observation that the formation of Pse-chr7 involved six syntenic segments and two centromeric loci (Figures 5A and S29). Specifically, the ringed anc-chr7 tangled with anc-chr4 by double joining. The first join at the centromeres was supported by adjacent αPCEN168 and αPCEN169 cenSats (at a distance of 1.5 Mb) with a small number of αPCEN169 cenSats (19 copies) at the breakpoint, while the second join was through the breakpoints of segments B and D in anc-chr4 and the breakpoints of segments F and G in anc-chr7 (Figures 5E and S29; Table S12).

## DISCUSSION

To investigate the evolutionary history of centromeres and their roles in shaping karyotypes, we assembled four near-T2T *Papaver* genomes of varying chromosome numbers. We constructed

a comprehensive satellite library for each species, identified a diverse set of cenSats, and experimentally validated two representatives, αPCEN168 and PCEN338. By comparing syntenic chromosome pairs, we observed various levels of genomic rearrangement in the (peri-)centromere, representing different stages of the expansion and contraction of cenSat arrays. An interspecies analysis of syntenic conservation revealed that many chromosome rearrangements were closely associated with the centromeres, suggesting a critical role for the latter in shaping karyotypes.

How has CMCR contributed to *Papaver* speciation? We observed that CMCRs between two subgenomes (SG3 and SG4) of the allopolyploid *P. setigerum* resulted in the loss of three centromeres (in particular the αPCEN169 cenSat from SG3) and the retention of the αPCEN168 cenSat on all seven centromeres of SG4 (Figure 3D). To explain the observation of CMCR with imbalanced sets of satellites in *Papaver*, we propose a model for centromere evolution incorporating both allopolyploidization and centromere drive (Figure 6). The rapid innovation of novel traits can follow the hybridization of two species, as this process essentially doubles the number of functional elements in the genome.[36,37] In this case, after hybridization, two sets of centromeric satellites (and their associated CENH3 proteins) will coexist in the same nucleus. Centromere drive (that is, the differential segregation potential of CENH3-satellite combinations[1]), in conjunction with the relative expression of the *CENH3* genes (red in Figure 6), can enhance the imbalance, leading over time to the preferential accumulation of one set (red in Figure 6), with the concomitant loss of either the entire or partial subgenome of the other set (blue in Figure 6). Compounding this, CMCRs may bring two different, and competing, satellites onto the same chromosome (while at the same time also bringing genetic material from different species, facilitating selection for novel traits). In this respect, CMCRs may create a ''battleground'' upon which different sets of cenSats and CENH3 proteins may act. The ''defeated'' *CENH3* gene (the blue one in Figure 6) may be downregulated or lost from the population, while

the "winning" satellite will become fixed in the population and homogenized.

Our observations of distinct cenSat patterns in *P. setigerum* and *P. somniferum* suggest a process of "homogenizing" after hybridization (Figure S30A). For instance, although *P. somniferum* contains SG1 and SG2, only one subgenome expressed *CENH3*, and only one principal cenSat (PCEN338) was observed (Figures 3E and S30B), suggesting an extended period of competition among satellites in which one has become dominant above all others (Figure S30A). In SG3 and SG4 of *P. setigerum*, both satellites remain, with αPCEN168 the most abundant cenSat in those chromosomes (chr6 and 7) generated from CMCR between SG3 and SG4 (Figures 3D, S25, and S29); this suggests an "ongoing battle" after hybridization between the cenSats of SG3 and SG4 (Figure S30A). αPCEN168 has the potential to "win," as evidenced by two observations. Firstly, we observe the coexistence of αPCEN169 (small array with 1,416 copies) and αPCEN168 (large array with 3,460 copies) in *P. setigerum* chr7 (Figure S29; Table S12), and secondly, the sequence similarities among repeat units in the αPCEN168 array (mean value, 84.0%) are significantly higher than that of the αPCEN169 array (mean value, 81.8%) ($p < 2.2E-16$, two-sided Wilcoxon rank-sum test) (Figure S29D). Consistently, the *CENH3* gene on SG4 is still expressed, while its counterpart on SG3 is lost (Figure S30B). Moreover, only one rearrangement was observed between the SG1-SG2 group and SG3-SG4 group (Figure 3D), suggesting that cross-subgenome group rearrangement and satellite competition after the second hybridization may be in its initial stage (Figure S30A). In addition, we investigated the sequence diversity of *CENH3* in *Papaver* species. We first identified the *CENH3* genes as *Pbr03G53170.1* in *P. bracteatum* by leveraging the syntenic relationship with *PsoCENH3*. We then determined the amino acid sequence identity for syntenic gene pairs between *P. bracteatum* and *P. somniferum*, as well as between *P. bracteatum* and *P. setigerum* (Figure S30C). Our analysis revealed that the sequence identity of *PsoCENH3*-related genes is situated to the left of the peaks, suggesting a higher level of amino acid diversity in CENH3 compared to other genes.

In summary, by characterizing the centromeric satellome of four *Papaver* species, we illustrate the interplay between the opposing forces (with respect to cenSat diversity) of centromere drive and hybridization and highlight the complex and nuanced role the centromere plays in speciation. The global distribution of *Papaver* species and their variety of karyotypes make them a valuable genetic resource; hence, a pan-*Papaver* genome project based on high-quality genome assemblies and epigenetic data from multiple accessions has the potential to refine this model, providing new insights into our understanding of centromere evolution and its role in speciation.

### Limitations of the study
Precise identification of cenSats necessitates near-T2T genome assemblies, which remain elusive for non-model species and pose a challenge for the application of our cenSat identification pipeline. Our cross-comparison model is designed for the complex evolutionary history of the *Papaver* genus, which comprises subgenome evolution, (competing) satellite-type centromeres,

and two rounds of allopolyploidization, with its associated genome rearrangements. Our model would apply to species from the same genus and with sufficient synteny, but given the complexity of this evolutionary scenario, we would approach other species with caution. In particular, this model would not be readily applicable to other types of centromeres, such as point centromere (e.g., budding yeast[38]), transposon-based centromeres (e.g., in green algae[39]), and holocentromeres (e.g., in *Rhynchospora* species[40]). Furthermore, DNA methylation is an important factor in centromere study. Due to the current limitation of our data, it is hard for us to further investigate the DNA methylation patterns for characterizing centromeres in *Papaver* species.

### STAR★METHODS

Detailed methods are provided in the online version of this paper and include the following:

- KEY RESOURCES TABLE
- RESOURCE AVAILABILITY
  - Lead contact
  - Materials availability
  - Data and code availability
- EXPERIMENTAL MODEL AND STUDY PARTICIPANT DETAILS
  - Plant materials
- METHOD DETAILS
  - Karyotyping of *P. bracteatum*
  - HiFi long-read sequencing
  - Oxford Nanopore ultra-long sequencing
  - Illumina paired-end read sequencing
  - Hi-C sequencing
  - RNA sequencing
  - Genome size estimation for *P. bracteatum*
  - Genome assembly
  - Assembly evaluation
  - Genome annotation
  - Phylogenomic analysis
  - Gene expression analysis
  - Satellite library construction
  - Centromere satellite identification
  - Cross-species cenSat array comparison
  - Higher-order repeat (HOR) annotation
  - CENH3 antibody generation
  - ChIP-seq
  - ChIP-seq data analysis
  - Chromosome preparation and FISH
  - Evolutionary history reconstruction
  - Reconstruction of cenSat evolution trajectory
- QUANTIFICATION AND STATISTICAL ANALYSIS

### SUPPLEMENTAL INFORMATION

### ACKNOWLEDGMENTS

We thank Ian A. Graham, Zemin Ning, Wen Wang, Zongcheng Lin, and Guojie Zhang for helpful discussions on genome analysis and Jing Hai, Huanhuan Zhao, Chen Huang, and Dongdong Tong for administrative and technical support. We thank Miss Ying Hao at the instrument analysis center of Xi'an Jiaotong University for her assistance with the usage of the Leica TCS SP8 STED 3X. This study was supported by the National Natural Science Foundation of China (32125009, 62172325, 32070663, 32200510, and 82304671), the

National Key R&D Program of China (2022YFC3400300), the Natural Science Foundation of Shaanxi Province (2024JC-JCQN-28), and the Fundamental Research Funds for the Central Universities (xzy012024088).

## AUTHOR CONTRIBUTIONS

K.Y. and X.Y. designed and supervised research. Y.J. collected materials for sequencing. X.Y., S.G., H.G., S.W., Y.Z., and T.X. performed the genome assembly, assembly evaluation, genome annotation, synteny analysis, ChIP-seq, and RNA-seq analysis. X.Y. and S.G. predicted and validated the cenSats. Y.J. performed experimental validation of the cenSats. B.W. performed phylogenomic analysis. S.G., X.Y., and H.G. performed ancestral genome reconstruction. Y.J. performed the karyotyping of *P. bracteatum*. K.Y., X.Y., and S.G. interpreted the dynamic evolution of centromeres and chromosomes after hybridization. X.Y., and S.G. prepared the figures and tables. K.Y., X.Y., S.G., and S.J.B. wrote the paper. All authors read and approved the final manuscript.

## DECLARATION OF INTERESTS

The authors declare no competing interests.

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

## STAR★METHODS

### KEY RESOURCES TABLE

| REAGENT or RESOURCE | SOURCE | IDENTIFIER |
|---|---|---|
| **Antibodies** | | |
| CENH3 in *P. rhoeas* | This paper | PrhCENH3; Cat#PHY6567A, RRID:AB_3107190 |
| CENH3 in *P. somniferum* | This paper | PsoCENH3; Cat#PHY6415A; RRID:AB_3107189 |
| **Biological samples** | | |
| *P. somniferum* | This paper | N/A |
| *P. bracteatum* | This paper | N/A |
| *P. setigerum* | This paper | N/A |
| *P. rhoeas* | This paper | N/A |
| **Peptides** | | |
| MQKDWQLARRLGGRGQYC | This paper | N/A |
| SDAGGKKRSYRHKPGAKC | This paper | N/A |
| **Critical commercial assays** | | |
| Genomic Kit for HiFi sequencing | GrandOmics | N/A |
| SMRTbell Express Template Prep Kit 2.0 | Pacific Biosciences | N/A |
| SMRTbell Enzyme Cleanup Kit | Pacific Biosciences | N/A |
| SMRTbell Enzyme Cleanup Kit | Pacific Biosciences | N/A |
| Sequel II Binding Kit 2.2 | Pacific Biosciences | N/A |
| Genomic DNA Kit for ONT sequencing | GrandOmics | N/A |
| BAC-long DNA Kit | GrandOmics | N/A |
| Ligation Sequencing 1D Kit (#SQK-LSK109) | Oxford Nanopore Technologies | N/A |
| Qubit DNA Assay Kit | Illumina | N/A |
| TruSeq Nano DNA HT Sample Preparation Kit | Illumina | N/A |
| QIAamp DNA Mini Kit | Qiagen | N/A |
| Qubit® RNA Assay Kit | Life Technologies | N/A |
| RNA Nano 6000 Assay Kit | Agilent Technologies | N/A |
| TruSeq RNA Library Preparation Kit | Illumina | N/A |
| TruSeq PE Cluster Kit v3-cBot-HS | Illumina | N/A |
| **Deposited data** | | |
| *P. somniferum* genome assembly | This paper | National Genomic Data Center (https://bigd.big.ac.cn/gwh/): GWHAZPI00000000.1 |
| *P. bracteatum* genome assembly | This paper | National Genomic Data Center (https://bigd.big.ac.cn/gwh/): GWHDRIO00000000 |
| *P. setigerum* genome assembly | This paper | National Genomic Data Center (https://bigd.big.ac.cn/gwh/): GWHAZPH00000000.1 |
| *P. rhoeas* genome assembly | This paper | National Genomic Data Center (https://bigd.big.ac.cn/gwh/): GWHAZPI00000000.1 |
| HiFi sequencing data of *P. somniferum* | This paper | National Genomic Data Center (https://bigd.big.ac.cn/gsa/): CRA012138 |
| HiFi sequencing data of *P. bracteatum* | This paper | National Genomic Data Center (https://bigd.big.ac.cn/gsa/): CRA012138 |
| HiFi sequencing data of *P. setigerum* | This paper | National Genomic Data Center (https://bigd.big.ac.cn/gsa/): CRA012138 |
| HiFi sequencing data of *P. rhoeas* | This paper | National Genomic Data Center (https://bigd.big.ac.cn/gsa/): CRA012138 |

*Continued*

| REAGENT or RESOURCE | SOURCE | IDENTIFIER |
| --- | --- | --- |
| ONT sequencing data of *P. somniferum* | This paper | National Genomic Data Center (https://bigd.big.ac.cn/gsa/): CRA012138 |
| PrhCENH3 ChIP-seq data of *P. rhoeas* | This paper | National Genomic Data Center (https://bigd.big.ac.cn/gsa/): CRA016610 |
| PsoCENH3 ChIP-seq data of *P. somniferum* | This paper | National Genomic Data Center (https://bigd.big.ac.cn/gsa/): CRA012138 |
| Illumina paired-end sequencing data of *P. bracteatum* | This paper | National Genomic Data Center (https://bigd.big.ac.cn/gsa/): CRA012138 |
| Hi-C sequencing data of *P. bracteatum* | This paper | National Genomic Data Center (https://bigd.big.ac.cn/gsa/): CRA012138 |
| RNA-seq data of *P. bracteatum* | This paper | National Genomic Data Center (https://bigd.big.ac.cn/gsa/): CRA012138 |
| *P. somniferum* genome assembly in 2021 | Yang et al.[19] | National Genomic Data Center (https://bigd.big.ac.cn/gwh/): PRJCA004217 |
| *P. bracteatum* genome assembly in 2022 | Catania et al.[23] | National Center for Biotechnology Information (https://www.ncbi.nlm.nih.gov/assembly/): PRJNA770669 |
| *P. setigerum* genome assembly in 2021 | Yang et al.[19] | National Genomic Data Center (https://bigd.big.ac.cn/gwh/): PRJCA004217 |
| *P. rhoeas* genome assembly in 2021 | Yang et al.[19] | National Genomic Data Center (https://bigd.big.ac.cn/gwh/): PRJCA004217 |
| ONT sequencing data of *P. setigerum* | Yang et al.[19] | National Genomic Data Center (https://bigd.big.ac.cn/gsa/): PRJCA004217 |
| ONT sequencing data of *P. rhoeas* | Yang et al.[19] | National Genomic Data Center (https://bigd.big.ac.cn/gsa/): PRJCA004217 |
| Illumina paired-end sequencing data of *P. somniferum* | Guo et al.[22] | National Center for Biotechnology Information (https://www.ncbi.nlm.nih.gov/sra): PRJNA435796 |
| Illumina paired-end sequencing data of *P. setigerum* | Yang et al.[19] | National Genomic Data Center (https://bigd.big.ac.cn/gsa/): PRJCA004217 |
| Illumina paired-end sequencing data of *P. rhoeas* | Yang et al.[19] | National Genomic Data Center (https://bigd.big.ac.cn/gsa/): PRJCA004217 |
| Hi-C sequencing data of *P. somniferum* | Yang et al.[19] | National Genomic Data Center (https://bigd.big.ac.cn/gsa/): PRJCA004217 |
| Hi-C sequencing data of *P. setigerum* | Yang et al.[19] | National Genomic Data Center (https://bigd.big.ac.cn/gsa/): PRJCA004217 |
| Hi-C sequencing data of *P. rhoeas* | Yang et al.[19] | National Genomic Data Center (https://bigd.big.ac.cn/gsa/): PRJCA004217 |
| RNA-seq data of *P. somniferum* | Guo et al.[22] | Gene Expression Omnibus (https://www.ncbi.nlm.nih.gov/geo/): GSE111119 |
| RNA-seq data of *P. setigerum* | Yang et al.[19] | National Genomic Data Center (https://bigd.big.ac.cn/gsa/): PRJCA004217 |
| RNA-seq data of seven seedling development stage of *P. sominiferum* | Li et al.[41] | National Genomic Data Center (https://bigd.big.ac.cn/gsa/): PRJNA508405 |
| **Software and algorithms** | | |
| SMRTLink v7.0 | N/A | https://www.pacb.com/support/software-downloads/ |
| Guppy v3.2.2+9fe0a78 | N/A | https://github.com/LernerLab/GuPPy |
| Lander Waterman algorithm | Lander et al.[42] | N/A |
| hifiasm v0.16.1 | Cheng et al.[43] | https://github.com/chhylp123/hifiasm |
| Flye v2.7-b1585 | Kolmogorov et al.[44] | https://github.com/fenderglass/Flye |
| HiCanu (v2.1.1) | Nurk et al.[45] | https://github.com/marbl/canu |
| Shasta (v0.10) | Shafin et al.[46] | https://github.com/paoloshasta/shasta |
| 3days-DNA (v180419) | Dudchenko et al.[47] | https://github.com/aidenlab/3d-dna |

*(Continued on next page)*

*Continued*

| REAGENT or RESOURCE | SOURCE | IDENTIFIER |
| --- | --- | --- |
| Ragtag (v2.0.1) | Alonge et al.[48] | https://github.com/malonge/RagTag |
| Juicer v1.5.7 | Durand et al.[49] | https://github.com/aidenlab/juicer |
| purge_dups v1.0.0 | Guan et al.[50] | https://github.com/dfguan/purge_dups |
| BUSCO (v5) | Manni et al.[24] | https://busco.ezlab.org/ |
| Merqury (v1.3) | Rhie et al.[51] | https://github.com/marbl/merqury |
| minimap2 (v2.24) | Li[52,53] | https://github.com/lh3/minimap2 |
| igvtools (v2.15.4) | Thorvaldsdottir et al.[54] | https://igv.org/ |
| VerityMap (v2.0.0) | Mikheenko et al.[29] | https://github.com/ablab/VerityMap |
| RepeatModeler vopen-1.0.8 | N/A | https://www.repeatmasker.org/RepeatModeler/ |
| RepeatMasker vopen-4.0.7 | N/A | https://www.repeatmasker.org/ |
| LTR_Finder v1.1 | Xu et al.[55] | https://github.com/xzhub/LTR_Finder |
| LTRHarvest v1.5.9 | Ellinghaus et al.[56] | http://genometools.org/ |
| LTR_retriever v2.8.5 | Ou et al.[57] | https://github.com/oushujun/LTR_retriever |
| TRF v4.09 | Benson[26] | https://tandem.bu.edu/trf/trf.html |
| Infernal (v1.1.2) | Nawrocki[58] | http://eddylab.org/infernal/ |
| MAKER2 pipeline (v2.31.8) | Campbell et al.[59] | https://www.yandell-lab.org/software/maker.html |
| AUGUSTUS (v3.3) | Keller et al.[60] | https://bioinf.uni-greifswald.de/augustus/ |
| SNAP (v2006-07-28) | Korf[61] | https://github.com/KorfLab/SNAP |
| GeneMark_ES (v3.48) | Lomsadze et al.[62] | http://exon.gatech.edu/GeneMark/ |
| Trinity (v2.1.1) | Grabherr et al.[63] | https://github.com/trinityrnaseq/trinityrnaseq |
| InterProScan (v5.25–64.0) | Jones et al.[25] | https://www.ebi.ac.uk/interpro/search/sequence/ |
| OrthoFinder v.2.3.4 | Emms[64] | https://github.com/davidemms/OrthoFinder |
| MAFFT (v7) | Katoh et al.[65] | https://mafft.cbrc.jp/alignment/software/ |
| Gblocks (v0.91b) | Castresana[66] | https://www.biologiaevolutiva.org/jcastresana/Gblocks.html |
| RAxML (v8.2.12) | Stamatakis[67] | https://github.com/amkozlov/raxml-ng |
| r8s v1.8 | Sanderson[68] | https://sourceforge.net/projects/r8s/ |
| Subphaser v1.2.5 | Jia et al.[31] | https://github.com/zhangrengang/SubPhaser |
| Hisat2 (v2.2.1) | Pertea et al.[69] | https://daehwankimlab.github.io/hisat2/ |
| Stringtie (v2.1.4) | Pertea et al.[69] | https://ccb.jhu.edu/software/stringtie/ |
| Lastz v1.04.00 | Harris[70] | https://www.bx.psu.edu/~rsharris/lastz/ |
| StringDecomposer v1.1.2 | Dvorkina et al.[71] | https://github.com/ablab/stringdecomposer |
| Louvain algorithm | Blondel et al.[27] Traag et al.[28] | N/A |
| HiCAT v1.1 | Gao et al.[30] | https://github.com/xjtu-omics/HiCAT |
| CLUSTAL Omega v1.2.4 | N/A | https://www.ebi.ac.uk/Tools/msa/clustalo/ |
| bowtie2 v2.5.0 | Langmead et al.[72] | https://github.com/BenLangmead/bowtie2 |
| DRIMM-Synteny | Pham et al.[73] | https://github.com/xjtu-omics/processDrimm/tree/master/drimm/drimm |
| IAGS | Gao et al.[32] | https://github.com/xjtu-omics/IAGS |
| Other | | |
| Custom scripts for build satellite library | This paper | https://github.com/xjtu-omics/BSLtool, https://doi.org/10.5281/zenodo.12179368 |
| *Papaver* genome annotation | This paper | https://xjtu-omics.github.io/Papaver-Genomics/, https://doi.org/10.5281/zenodo.12179378 |

## RESOURCE AVAILABILITY

### Lead contact

Further information and requests for resources and reagents should be directed to and will be fulfilled by the lead contact, Kai Ye (kaiye@xjtu.edu.cn).

CellPress

## Materials availability

This study did not generate new unique reagents.

## Data and code availability

- Raw HiFi, Oxford Nanopore, Illumina paired-end, Hi-C, RNA-seq, and ChIP-seq data generated in this study have been deposited at the National Genomics Data Center (https://ngdc.cncb.ac.cn/) Genome Sequence Archive and accession numbers are listed in the key resource table. The genome assembly data have been deposited at the Genome Warehouse of the National Genomics Data Center and accession numbers are listed in the key resource table. Annotations of the four genomes have been deposited in the *Papaver* Genomic Database, DOIs are listed in the key resource table.
- All original code has been deposited at Zenodo and is publicly available as of the date of publication. DOIs are listed in the key resources table.
- Any additional information required to reanalyze the data reported in this paper is available from the lead contact upon request.

## EXPERIMENTAL MODEL AND STUDY PARTICIPANT DETAILS

### Plant materials

To provide raw material for both genome and transcriptome sequencing, *P. setigerum* variety DCW1, *P. rhoeas* variety YMR1, *P. bracteatum* variety PBR1, and *P. somniferum* variety HN1 were grown in Azalea pots in a regulated growth chamber with 16 h of light, located at the Xi'an Jiaotong University Laboratory of BioData Sciences. The growth substrate was a soil mix of four parts potting mix, two parts natural soil and one-part Vermiculite. For long-read genome sequencing and chromatin conformation capture (Hi-C) sequencing, fresh leaves (the four uppermost ones) were harvested from six-week-old seedlings of each of the four species. For transcriptome sequencing, material was sampled from other leaves of the same sample. All materials were collected, rinsed with water and surface-sterilized with 70% ethanol for 10 min to remove commensal contaminants before being processed for library construction and sequencing.

## METHOD DETAILS

### Karyotyping of *P. bracteatum*

*P. bracteatum* seeds were washed and placed in a culture dish with moist filter paper in an incubator at 25°C to allow germination until the root grew to about 1 cm. For karyotyping, about 0.5 cm fresh root tips were cut off in the morning, and immediately placed in a 0.004 M 8-hydroxyquinoline solution for 4 h in darkness, at room temperature. The root tips were then fixed in Carnoy's fluid (absolute ethanol: acetic acid = 3: 1 V/V) overnight, and stored in 70% ethanol at 4°C for further studies. In order to achieve optimal separation of the chromosomes at metaphase, the root tips were thoroughly washed with distilled water, and then macerated in 1 M HCl for 9 min at 60°C for acid hydrolysis. After dissociation, the root tips were placed in distilled water for 15 min to ensure hypotension, then the root tips were stained by improved carbol-fuchsin solution for 10 min, and squashed on a glass slide. Finally, chromosomes were examined with a microscope (Olympus CX23, Japan) and photographs were taken. Photographs were processed using Adobe Photoshop 7.0 (Adobe Systems, San Jose, USA). The karyotyping results are shown in Figure S1 and confirm the karyotype of *P. bracteatum* as $2n$ = 14. We repeated the karyotyping experiment three independent times with identical results (Figure S1).

### HiFi long-read sequencing

Leaves of the four *Papaver* species were collected separately. For each sample, high molecular weight genomic DNA was prepared by the CTAB method and followed by purification with the GrandOmics Genomic Kit (Wuhan, China) for regular sequencing, according to the manufacturer's standard operating procedure. DNA degradation and contamination of the extracted DNA was monitored using 1% agarose gels. DNA purity was quantified using the NanoDrop One UV-Vis spectrophotometer (Thermo Fisher Scientific, USA), of which OD260/280 ranged from 1.8 to 2.0 and OD 260/230 from 2.0 to 2.2. DNA concentration was measured using a Qubit 4.0 Fluorometer (Invitrogen, USA). For each sample, SMRTbell target size libraries were constructed for sequencing according to PacBio's standard protocol (Pacific Biosciences, CA, USA) using 15kb preparation solutions. The main steps for library preparation include: (1) gDNA shearing, (2) DNA damage repair, end repair and A-tailing, (3) ligation with hairpin adapters from the SMRTbell Express Template Prep Kit 2.0 (Pacific Biosciences), (4) nuclease treatment of SMRTbell library with SMRTbell Enzyme Cleanup Kit, (5) size selection, and (6) binding to polymerase. In brief, a total amount of 8 μg DNA was used for the DNA library construction. The genomic DNA sample was sheared by g-TUBEs (Covaris, USA) according to the expected size of the fragments for the library. Single-strand overhangs were then removed, and DNA fragments were damage-repaired, end-repaired and A-tailed. Fragments were then ligated with the hairpin adapter for PacBio sequencing. The library was then treated by nuclease with the SMRTbell Enzyme Cleanup Kit and purified using AMPure PB Beads, with the Agilent 2100 Bioanalyzer (Agilent technologies, USA) used to detect the size of library fragments. Target fragments were screened by the PippinHT (Sage Science, USA). Sequencing was performed by GrandOmics (Wuhan, China) using a PacBio Sequel II with the Sequencing Primer V5 and Sequel II Binding Kit 2.2. SMRTLink v7.0 was used to generate the HiFi reads from the sequenced raw subreads using the 'css' command with parameters ''–min-passes 1 –min-rq 0.99 –min-length 100''.

### Oxford Nanopore ultra-long sequencing

Fresh leaves from *P. somniferum* were collected and used for DNA extraction. DNA was extracted using the GrandOmics Genomic DNA Kit and BAC-long DNA Kit, following the manufacturer's instructions for Oxford Nanopore (ONT) ultra-long library construction. The quality and quantity of total DNA was evaluated using a NanoDrop One UV–Vis spectrophotometer (ThermoFisher Scientific, Waltham, MA) and Qubit 3.0 Fluorometer (Invitrogen life Technologies, Carlsbad, CA), respectively. The Blue Pippin system (Sage Science, Beverly, MA) was used to retrieve large DNA fragments by gel cutting. For the ultra-long Nanopore library, approximately 8–10 μg of genomic DNA was selected (>50 kb) with the SageHLS HMW library system (Sage Science), and then processed using the Ligation Sequencing 1D Kit (Catalog No. SQK-LSK109, Oxford Nanopore Technologies, Oxford, UK) according to the manufacturer's instructions. DNA libraries (approximately 400 ng) were constructed and sequenced on the PromethION (Oxford Nanopore Technologies) at the Genome Center of GrandOmics (Wuhan, China). Base calling was performed using Guppy v3.2.2+9fe0a78 with parameter "–c dna_r9.4.1_450bps_fast.cfg".

### Illumina paired-end read sequencing

Fresh leaves from *P. bracteatum* were collected and used for DNA extraction. DNA purity was checked using the NanoPhotometer spectrophotometer (IMPLEN, CA, USA). DNA concentration was measured using a Qubit 2.0 Flurometer (Life Technologies, CA, USA) with the Qubit DNA Assay Kit. A total amount of 1.5μg DNA was used as input material for the DNA sample preparations. Sequencing libraries were generated using the TruSeq Nano DNA HT Sample Preparation Kit (Illumina USA) following the manufacturer's instructions, with index codes added to each sample. Briefly, the DNA sample was fragmented by sonication to a size of 350bp, then DNA fragments were end-polished, A-tailed, and ligated with the full-length adapter for Illumina sequencing with further PCR amplification. PCR products were then purified (using the AMPure XP system) and the libraries analyzed for size distribution using the Agilent2100 Bioanalyzer and quantified using real-time PCR. Finally, the libraries were sequenced using the Illumina NovaSeq 6000 platform, generating 150bp paired-end reads with an insert size of approximately 350bp. Raw sequencing reads were processed using in-house C scripts to apply a series of quality control (QC) procedures: (1) Removing reads with ≥10% un-identified (N) nucleotides; (2) Removing reads where >50% of bases had Phred quality <5; (3) Removing reads where >10 nt aligned to the adapter, allowing ≤10% mismatches; (4) Removing read pairs when read 1 and read 2 were identical (as these are likely to be PCR duplicates).

### Hi-C sequencing

About 2 g of *P. bracteatum* material (fresh leaves) was cut into 1 to 2 mm strips, which were fixed with 2% final concentration fresh formaldehyde in NIB buffer (20 mM HEPES, pH 8.0, 250 mM sucrose, 1 mM MgCl2,5mM KCl, 40% (v/v) glycerol, 0.25% (v/v) Triton X-100, 0.1 mM PMSF, and 0.1% (v/v) β-mercaptoethanol) at 4°C for 45 min in a vacuum. Formaldehyde was added at a final concentration of 0.375 M glycine under vacuum infiltration for an additional 5 min. The samples were washed twice in ice-cold water. The clean samples were frozen in liquid nitrogen and then ground to a powder and resuspended in the NIB buffer. The solution was then filtered through one layer of Miracloth. The nuclei isolated from these tissues were lysed with 0.1% (w/v) final concentration SDS at 65°C for 10 min and then SDS molecules were added using Triton X-100 at a 1% (v/v) final concentration. The DNA in the nuclei was then digested by adding 200U DpnII (NEB) and incubating the samples at 37°C for 2 h. Restriction fragment ends were labeled with biotinylated cytosine nucleotides by Biotin-14-dATP (TriLINK). Blunt-end ligation was carried out at 16°C overnight in the presence of 50 Weiss units of T4 DNA ligase. After ligation, the cross-linking was reversed by 200 μg/mL proteinase K (Thermo) at 65°C overnight. DNA purification was achieved through QIAamp DNA Mini Kit (Qiagen) according to the manufacturer's instructions. Purified DNA was sheared to a length of ∼400 bp. Finally, the Hi-C libraries were quantified and sequenced using the Illumina NovaSeq 6000 platform.

### RNA sequencing

RNA degradation and contamination was monitored on 1% agarose gels. RNA concentration was measured using a Qubit 2.0 Flurometer (Life Technologies, CA, USA) with the Qubit RNA Assay Kit. RNA integrity was assessed using the RNA Nano 6000 Assay Kit with the Bioanalyzer 2100 system (Agilent Technologies, CA, USA). A total amount of 1 μg RNA per sample was used as input material for the RNA sample preparations. Sequencing libraries were generated using the TruSeq RNA Library Preparation Kit (Illumina, USA) following manufacturer's recommendations and index codes were added to attribute sequences to each sample. Briefly, mRNA was purified from total RNA using poly-T oligo-attached magnetic beads. First strand cDNA was synthesized using random hexamer primer and M-MuLV Reverse Transcriptase (RNase H-). Second strand cDNA synthesis was subsequently performed using DNA Polymerase I and RNase H. The remaining overhangs were converted into blunt ends via exonuclease/polymerase activity. After adenylation of the 3′ ends of DNA fragments, Illumina adaptor was ligated to prepare for hybridization. In order to select cDNA fragments of length 150–200 bp, library fragments were purified with the AMPure XP system (Beckman Coulter, Beverly, USA). PCR was then performed with the Phusion High-Fidelity DNA polymerase, universal PCR primers and index (X) Primer. Finally, PCR products were purified (AMPure XP system) and library quality assessed using the Agilent Bioanalyzer 2100 system. The clustering of the index-coded samples was performed on a cBot Cluster Generation System using TruSeq PE Cluster Kit v3-cBot-HS (Illumina) according to the

manufacturer's instructions. After generating clusters, libraries were sequenced on an Illumina NovaSeq 6000, generating 150 bp paired-end reads.

### Genome size estimation for *P. bracteatum*

For *P. bracteatum*, we estimated genome size $G$ based on whole genome Illumina paired-end sequencing data using k-mer frequency analysis with $k = 17$ following the Lander Waterman algorithm[42]: $G = K_{num}/K_{depth}$, where $K_{num}$ denotes the number of k-mers, and $K_{depth}$ the k-mer depth.[42] For *P. bracteatum*, we obtained 89,085,253,385 kmers, at a depth of 35 (Figure S3). Therefore, the estimated genome size is 2,545,292,953 bp.

### Genome assembly

We developed a genome assembly pipeline to *de novo* assemble the four *Papaver* genomes (Figure S2). Contigs were assembled using hifiasm (v0.16.1),[43] Flye (v2.7-b1585),[44] HiCanu (v2.1.1),[45] and Shasta (v0.10)[46] based on the sequenced HiFi reads and Oxford Nanopore reads. We scaffolded the genome using 3days-DNA (v180419)[47] and the Ragtag (v2.0.1)[48] 'scaffold' subcommand. The Ragtag 'patch' subcommand was applied to close the assembled gaps. Then, Hi-C data was aligned to the gap-closed scaffolds and the Hi-C interaction map constructed using Juicer (v1.5.7).[49] Based on the Hi-C map, we manually checked the genome and obtained the final scaffolds. The Oxford Nanopore reads of *P. rhoeas* and *P. setigerum*, and the Hi-C data of *P. somniferum*, *P. rhoeas* and *P. setigerum* were obtained from our previously published study.[19] It should be noted that we did not sequence ONT reads for *P. bracteatum*. The detailed procedure of the assembly pipeline is as follows.

Step 1. Contig generation.

```
#hifiasm for HiFi data
hifiasm -o $outsign -t $t $hifi_reads_list
gfatools gfa2fa $outsign.bp.p_ctg.gfa > $outsign.bp.p_ctg.fa.
#Flye for HiFi data
flye –pacbio-hifi $hifi_reads_list –genome-size $size –thread $t.
#HiCanu for HiFi data
canu -p $outsign genomeSize = $size maxThreads = $t minReadLength = 2000 minOverlapLength = 2000 useGrid = false -pacbio-hifi $hifi_reads_list
#Shasta for HiFi data
shasta-0.10 –input $hifi_reads_list –config HiFi-Oct2021.conf –threads $t 2> shasta.log.
#Shasta for ONT data
shasta-0.10 –input $ont_reads_list –config Nanopore-Sep2020.conf –threads $t 2> shasta.log.
```

For the HiCanu contigs, the assembled genome size was larger than the actual genome size, so we used purge_dups v1.0.0[50] to remove the redundant sequences.

Step 2. Generating scaffolds by 3days-DNA and "ragtag scaffold" command. We used 3days-DNA on the contigs from hifiasm (v0.16.1) as well as Hi-C data to produce the first version of scaffolds (scaffolds v1). We found that contig N50 was reduced after 3days-DNA scaffolding (Table S2), therefore, we rescaffolded the hifiasm contigs using "ragtag scaffold", considering the 3days-DNA scaffold as reference: "ragtag.py scaffold -r -o $outdir $3days-DNA_scaffold $hifiasm_contigs". This produced the second version of the scaffolds (scaffolds v2).

Step 3. Closing gaps in scaffolds v2 by using "ragtag patch" command. We iteratively performed gap closing on the scaffolds constructed in Step 2 using the contigs from Flye, HiCanu, Shasta (for HiFi reads), and Shasta (for ONT reads): "ragtag.py patch –aligner minimap2 -u -o $outdir $scaffold $contigs". After each round of gap-closing, we produced "scaffold v6" ("scaffold v5" for *P. bracteatum* since in this case there were no Shasta ONT contigs) (Table S2).

Step 4. Construction of a Hi-C contact map. We aligned Hi-C reads to scaffolds v6 and constructed the Hi-C map using Juicer with default parameters.

Step 5. Manually checking scaffold v6 to correct mis-assemblies based on the Hi-C contact map and removing duplicated sequences to produce a final set of scaffolds (Tables 1 and S2).

### Assembly evaluation

We first used Benchmarking Universal Single-Copy Orthologs (BUSCO) (v5)[24] with the embryophyta_odb10 dataset to evaluate the completeness of each of the four *Papaver* genomes (Tables 1 and S2). We then evaluated their genome quality using Merqury (v1.3)[51] based on short-read and HiFi sequencing data with parameter $k$ estimated as 21 (Table S2).

Finally, we aligned both the HiFi and ONT reads to the final assembled genomes by minimap2 (v2.24)[52,53] and investigated the read coverage by igvtools (v2.15.4)[54] to check potential mis-assemblies (Figure S18). VerityMap (v2.0.0)[29] was used to detect deviated reads in potential centromere region to evaluate the quality of centromere assemblies based on HiFi reads (Figure S19).

## Genome annotation

### Repetitive element annotation

Repbase[74] and the species-specific *de novo* constructed repeat library were used to annotate the repetitive elements in each of the four *Papaver* genomes. Repbase was downloaded from http://www.girinst.org/repbase/ on January 27[th], 2017, and the species-specific *de novo* repeat library was constructed using RepeatModeler (vopen-1.0.8. We used RepeatMasker (vopen-4.0.7 to annotate the repeat elements. LTR_Finder (v1.1),[55] LTRHarvest (v1.5.9)[56] and LTR_retriever (v2.8.5)[57] were used to detect long terminal repeat (LTR) elements. The detail of repetitive element annotation results showed in Table S4.

### Tandem repeat annotation

We used tandem repeat finder (TRF) (v4.09)[26] to detect tandem repeats in each genome. After detection, we calculated the total size of tandem repeat regions and the unit length distribution (Figure 1).

### Non-coding RNA annotation

We used the Infernal (v1.1.2) package[58] and the Rfam database (v14.1) (https://ftp.ebi.ac.uk/pub/databases/Rfam/CURRENT/)[75] to annotate non-coding RNAs (ncRNAs) in each of the four *Papaver* genomes. We indexed the Rfam database using the command 'cmpress Rfam.cm', and then predicted the ncRNAs using cmscan, as follows:

    cmscan -Z $genome_size –cut_ga –rfam –nohmmonly –tblout $out_sign.tblout –fmt 2 –clanin $RFAMDIR/Rfam.clanin –cpu $t $RFAMDIR/Rfam.cm $REF > $out_sign.cmscan

    grep -v ' = ' $out_sign.tblout >$out_sign.deoverlapped.tblout

We predicted 15,305, 11,092, 35,556, and 22,406 ncRNAs in *P. bracteatum*, *P. rhoeas*, *P. setigerum*, and *P. somniferum*, respectively (Table S4), and classified the ncRNAs (into, e.g., miRNA, snRNA, rRNA, and tRNA) using class information from http://rfam.xfam.org/search#tabview=tab4.

### Protein-coding gene prediction and functional annotation

Protein-coding genes were predicted using the MAKER2 pipeline (v2.31.8).[59] In short, MAKER2 first masked repetitive elements in the assembled genomes using RepeatMasker. Then, applies both evidence-based and *ab initio* gene predictors to predict protein-coding genes. For the evidence-based model, MAKER2 uses Blast to align protein and transcript sequence to the genome. The alignments were further polished by Exonerate to produce gene models.[76] MAKER2 then performs the *ab initio* gene prediction based on the assembly sequence and then compared the predicted gene models to those determined by transcript and protein alignment to revise the model predictions. The confidence of each gene model was then assessed using both Annotation Edit Distance (AED) and exonAED (eAED) method, which quantify the normalized distance between the gene model and its supporting evidence.

Three *ab initio* gene predictors were used: AUGUSTUS (v3.3),[60] SNAP (v2006-07-28)[61] and GeneMark_ES (v3.48).[62] Tomato (*Solanum lycopersicum*) was used as the species model for the AUGUSTUS gene prediction, and the pre-trained model of *Arabidopsis thaliana* used as input for the Hidden Markov Models of SNAP and GeneMark_ES. Protein sequences were obtained from Swiss-Prot (https://www.uniprot.org/downloads; downloaded in January 2020) and for *A. thaliana* (TAIR10),[77] *Beta vulgaris* (RefBeet-1.2.2)[78] and *Vitis vinifera* (12X),[79] from the Ensembl Plants database (http://plants.ensembl.org/index.html). Transcripts were *de novo* assembled by Trinity (v2.1.1)[63] using the RNA-seq data of four species.

We filtered these genes to produce high-confidence gene sets of 40,371, 42,133, 126,422, and 64,087 genes in *P. bracteatum*, *P. rhoeas*, *P. setigerum*, and *P. somniferum*, respectively (Tables 1 and S4) using the following filter criteria: **1)** excluding genes lacking either transcript or protein homology support, **2)** excluding genes with AED or eAED larger than 0.5, and **3)** excluding genes overlapping with annotated ncRNAs (non-coding RNAs). We functionally annotated the predicted protein-coding genes using InterProScan (v5.25–64.0) with default parameters.[25] In total, 65.94%, 65.98%, 64.21%, and 65.05% of the predicted genes in *P. bracteatum*, *P. rhoeas*, *P. setigerum*, and *P. somniferum*, respectively, could be annotated with functional domains (Table S4).

## Phylogenomic analysis

To investigate the evolutionary history of the four *Papaver* genomes, we conducted phylogenomic analysis alongside five other angiosperm species: *P. californicum*,[23] *P. nudicaule*,[23] *Corydalis tomentella*[80]*, Aquilegia coerulea*[81]*,* and *Macleaya cordata*.[82] Single-copy orthologs are commonly used to achieve robust phylogenetic reconstruction with high confidence and concordance. Applying OrthoFinder v.2.3.4[64] we detected 13 single-copy orthologs from nine angiosperm genomes (Table S5). To construct a phylogenetic tree, single-copy ortholog pairs were aligned with MAFFT (v7),[65] with the conserved sites in the alignments extracted using Gblocks (v0.91b)[66] with default parameters, followed by maximum likelihood phylogenomic tree construction using RAxML (v8.2.12)[67] with 100 bootstraps (Figure S9). The divergence times between species were estimated using r8s v1.8[68] with the penalized likelihood method and parameter 'setsmoothing = 1000', with an constrain taxon time of *Aquilegia-Papaver* (102.9–117.2 Mya), of *Macleaya–Papaver* (31.5–75.7 Mya), of *P. rhoeas-P. somniferum* (7.2–11.1 Mya) and of *Corydalis–Papaver* (65.0–110.5 Mya), obtained from TimeTree.[83] We estimated the divergence time of *P. somniferum* and *P. bracteatum* to be approximately 8.2 Mya (Figure S9), consistent with TimeTree.

To construct the subgenome tree, we employed Subphaser v1.2.5[31] to obtain subgenome coordinates and partitioned the genome accordingly (Figure 4). The phylogenomic analysis and divergence time estimation pipeline for the subgenomes remained the same as described above, utilizing the divergence time of *Corydalis–Papaver* (65.0–110.5 Mya) and *P. bracteatum-P. somniferum* (8.2 Mya).

## Gene expression analysis

The cleaned RNA reads were aligned to the assembled genome of each *Papaver* species using Hisat2 (v2.2.1)[69] with transcripts assembled and quantified using Stringtie (v2.1.4)[69] and Ballgown, respectively, with default parameters. Gene expression level was quantified as TPM (transcripts per million).

## Satellite library construction

To construct a satellite library for the four *Papaver* species, we first ran TRF[26] for each genome, identifying those tandem repeat (TR) regions where the repeat copy number was >100 and the length of the consensus repeat unit (*CRUseq*) was >100 bp (Figure S10). If the overlap of two repeat regions was >80%, we removed the repeat with the longer *CRUseq*. Since noise exists in TR regions, we performed StringDecomposer v1.1.2[71] for each region with *CRUseq* as the template, discarding those regions with identity <80%. We next built a TR network as follows, where each node was a TR region and the edges connecting them were denoted $TR_1$ and $TR_2$. First, we performed Lastz v1.04.00[70] on the whole genome using the *CRUseq* of $TR_1$ as the query sequence ($CRUseq_{TR_1}$), and then filtered the Lastz alignment regions with identity <80% or an alignment length <80% of the $CRUseq_{TR_1}$ length. If at least one of the filtered $CRUseq_{TR_1}$ Lastz regions overlapped with $TR_2$, we built an edge between $TR_1$ and $TR_2$. After network construction, we detected satellite communities using the Louvain algorithm.[27,28] For community $C_i$, we considered the node with the maximum degree as the representative TR region, denoted as $TR\_Rep_{C_i}$. Each community $C_i$ represents a satellite, and the satellite unit sequence was denoted by the *CRUseq* of $TR\_Rep_{C_i}$. We sorted satellites based on sequence AT content and named each satellite according to the template "species name abbreviation + sequence length + S + rank" (Table S6). The species name abbreviations are Prh (*P. rhoeas*), Pbr (*P. bracteatum*), Pso (*P. somniferum*), and Pse (*P. setigerum*). We restricted subsequent analysis to satellites with total genome size larger than 1Mb and AT content higher than 60%.

## Centromere satellite identification

Only one satellite Prh168S1 appeared in all *P. rhoeas* chromosomes with AT content higher than 60% and total genomic size larger than 1 Mb² (Table S6). We therefore defined Prh168S1 as the *P. rhoeas* cenSat and validated it by FISH (Figures 2A, S11A, and S11B). The other cenSats of the remaining *Papaver* species were identified based on syntenic relationships around Prh168S1 (Figures 2B, S14, and S15, and Table S8). First, we used Orthofinder[64] to pairwise detected orthogroups based on protein sequences between *P. rhoeas* and *P. somniferum*, *P. setigerum* and *P. bracteatum*, respectively. We assigned a unique ID for each orthogroup and transformed genomes into orthogroup ID sequences. Then, syntenic relationships were detected by running, in a pairwise fashion, Drimm-Synteny[73] with parameter of cycle length as 20 based on orthogroup ID sequences. To obtain the genome coordinates for each block in each species, we first obtained the orthogroup ID sequences for each block. We then determined the longest common subsequence between block orthogroup ID sequences and initial orthogroup ID sequences to obtain the start and end genes, thereby determining the block genomic coordinates. For each species, we identified centromere satellites based on both cenSats coordinates and the pericentromeric syntenic relationships with *P. rhoeas*. Where multiple satellites had a pericentromeric syntenic relationship, we selected the most abundant one as the centromere satellite (Table S9). For *P. setigerum*, *P. somniferum*, and *P. bracteatum*, results are visualized as Figures 2B, S14 and S15, respectively.

## Cross-species cenSat array comparison

For each satellite $Sat_i$, we performed Lastz[70] alignment on the four genomes considering the satellite unit sequence as query. We filtered alignment results with identity <80% or alignment length <80% of the satellite sequence length, and obtained the region se $Sat\_R_i$ (Table S12). For a satellite pair $Sat_i$ and $Sat_j$, we connected them when at least one region pair in $Sat\_R_i$ and $Sat\_R_j$ overlapped with >80% identity. After cross-species satellite similarity network construction, we detected communities using the Louvain algorithm[27,28] and defined the representative centromere satellites based on the results. (Figure 3A and Table S11).

## Higher-order repeat (HOR) annotation

We performed HOR annotation analysis of the four representative cenSat arrays in their corresponding species, e.g., PCEN338 in *P. somniferum* (Pso338S1), αPCEN168 in *P. rhoeas* (Prh168S1), αPCEN169 in *P. setigerum* (Pse169S11), and PCEN238 in *P. bracteatum* (Pbr238S1), by applying our previous published method HiCAT v1.1.[30] Two inputs are required for HiCAT: the template satellite sequence and the sequence requiring annotation. Taking Pso338S1 as an example, we merged adjacent regions (detected by Lastz) with distance less than 5 kb and filtered the merged regions less than 5 kb. For each remaining region, we obtained the sequence using samtools v1.9[84] and then performed HiCAT with "-i sequence_fa_file -t template_fa_file". All regions shared the same template sequence; that is, the consensus repeat unit of the representative TR region for Pso338S1 (Table S6). After annotation, we summarized HOR unit lengths based on *out_all_layer* files for all regions (Figure 3F).

## CENH3 antibody generation

We performed BlastP to search CENH3 from predicted protein-coding genes of *P. somniferum* based on two CENH3 protein sequences: HTR12 from *Arabidopsis thaliana* (UniProt Accession ID: QBRVQ9), and NnCenH3 from *Nelumbo nucifera*.[85] We obtained 56 and 55 hits for NnCenH3 and HTR12, respectively. We applied CLUSTAL Omega (1.2.4) (https://www.ebi.ac.uk/Tools/msa/clustalo/) to get the multiple sequence alignment and the representative alignment shown in Figure S16A. Based on the sequence

features used to distinguish CENH3 from canonical H3 histones in plant (H3: SAVA, H3.3 HAVL),[3] we detected four candidate CENH3 genes in *P. somniferum* (Figure S16B). We calculated the gene expression of these four candidate CENH3s based on seven different seedling stages[41] (Figure S16B), and found that *Pso04G02820.1* had the highest expression in all seedling stages, indicating it is the most probable gene encoding CENH3 in *P. somniferum*. We named this gene *PsoCENH3*. The sequence feature of the *PsoCENH3* histone fold domain is EALT. The amino acid sequence of PsoCENH3 is:

>PsoCENH3.
MARRKKHFAQRYTPGGRQPPPPTPPPPSAAGSSSDAGGKKRSYRHKPGAKALQEIRKLQKNIDLLLPRAPFVRIVKEITDNFSKEVNR
WQAEALTALQEATEAFLVNTFEDAQLCAIHAKRVTIMQKDWQLARRLGGRGHYGSQPW

Furthermore, we identified CENH3 gene in *P. rhoeas* by integrating the evidences from syntenic gene-pair and protein sequence alignments. We found that *Prh03G45160.1* and *Pso04G02820.1* is a syntenic gene pair, and that Prh03G45160.1 is the best hit in BlastP alignments for PsoCENH3, NnCenH3-A, and HTR12_ARATH (Figure S12). We named *Prh03G45160.1* as *PrhCENH3*. In addition, we found the histone fold domain of *PrhCENH3* is EALT, the same as that of *PsoCENH3*. The amino acid sequence of PrhCENH3 is:

>PrhCENH3.
MARRKHFAQRYPPGGRQPQPPPPPPPPPSSSSDAAAKKRPYKRKPGTKALQDIRKLQKSIDLLMPRAPFVRIVKEITDNFSKEVNRW
QAEALTALQEAAEAFLVGTFQDAQLCAIHAKRVTIMQKDWQLARRLGGRGQYGSQPW

After PsoCENH3 and PrhCENH3 identification, we designed and synthesized the corresponding peptide to match the unique sequence of PsoCENH3 (SDAGGKKRSYRHKPGAKC) and PrhCENH3 (MQKDWQLARRLGGRGQYC). We injected the peptides into rabbits to construct a polyclonal antibody (PhytoAB, San Jose, CA, USA).

### ChIP-seq
Shanghai Jiayin Biotechnology Co., Ltd performed ChIP assays according to the standard crosslinking ChIP protocol with modifications. The leaves (approximately 4g) of 3-week-old seedings were harvested and ground into powder in liquid nitrogen. The grinded tissue was fixed with 1% formaldehyde at room temperature for 15 min, followed by 0.125 M glycine for 5 min. The sample was then washed, resuspended in lysis buffer, and sonicated to generate fragments. After sonication, immunoprecipitation was performed with the constructed antibody of PsoCENH3 and PrhCENH3. Then, we washed the immunoprecipitated complex and extracted DNA. DNA was purified using the Universal DNA Purification Kit (#DP214). We constructed ChIP-seq library by using the ChIP-Seq DNA sample preparation kit (NEBNext UltraII DNA) according to the manufacturer's instructions. The extracted DNA was ligated to specific adaptors followed by deep sequencing on an Illumina Novaseq 6000 using 150bp paired-end mode.

### ChIP-seq data analysis
We removed raw reads containing adapters, poly-N and with low-quality to generate the clean reads. We aligned both ChIP-seq and input paired-end clean reads of PsoCENH3 and PrhCENH3 using bowtie2 (2.5.0)[72] with parameters "–very-sensitive –no-mixed –no-discordant -k 10″" to the assembled *P. somniferum* and *P. rhoeas* genome, respectively. We converted the BAM files to tdf files using igvtools (2.15.4)[86] with parameters "count -z 5 -w 20000", and then obtained the coverage in BED file format using the "igvtools tdfto-bedgraph" and "igvtools bedgraphtobed" commands. We calculated the coverage ratio of ChIP to input (ChIP/input) and extracted the ChIP-seq coverage of centromeric and pericentromeric regions based on the location of the identified Pso338S1 and Prh168S1 arrays in *P. somniferum* and *P. rhoeas*, respectively (Table S7), and generated coverage figures (Figures 2D, S13, and S17) using IGV (2.15.4). Furthermore, we calculated the fold change of mean ChIP/input in (peri-)centromeric to other genomic regions to indicate the PsoCENH3 and PrhCENH3 ChIP-seq enrichment (Figures S13 and S17).

### Chromosome preparation and FISH
Chromosome preparation and FISH were performed as described in Ribeiro et al.[87] and Li et al.[88] with some modifications. Briefly, about 0.4–0.6 mm root tips were obtained from germinated seeds, and transferred quickly to 2 mM 8-hydroxyquinoline for pretreating, then fixed with 3:1 ethanol/acetic acid (V/V) solution for 24 h, and washed twice with 70% ethanol. The treated roots were digested in enzyme solution containing 2% (w/w) cellulose Onozuka R 10 (Yakult Pharmaceutical, Tokyo, Japan) and 1% (w/w) pectinase Y23 (Yakult Pharmaceutical, Tokyo, Japan) for 1 h at 37°C. The fully enzymatically digested root meristematic tissues were mashed with needles, and spread out over the slide with the help of acetic acid. After air drying, selected slides were UV-crosslinked for 2 min and stored at −20°C for FISH analysis.

50 bp nucleotides was selected from Prh168S1 and Pso338S1, and labeled directly with Alexa Fluor 488-5-dUTP (Thermo Fisher) for using as FISH probe. 6 μL of probe solution (100 ng/μL of probe in 2 × saline sodium citrate and 1 × Tris–ethylenediaminetetraacetic acid buffer) was added to each slide, heated for 5 min at 100°C, and left overnight at 55°C in a humid chamber. Slides were washed in 2 × saline sodium citrate for 20 min at 55°C, air-dried and mounted with Vectashield mounting medium (Vector Laboratories). FISH signals were captured using Leica TCS SP8 STED 3X, with images pseudo-colored and processed using Leica LAS X offline software. The FISH results showed in Figures 2A–2C, and S11.

The probe sequences were as follows.
>Prh168S1_probe
Gacatctattaatacttcaattttagaggaccgagagttattgataacaa
>Pso338S1_probe
Tcaagatagatgaaatttgttaggatttggattagttctgcatggtcttt

## Evolutionary history reconstruction

Evolutionary history reconstruction included four steps: detecting orthogroups, building syntenic blocks, inferring chromosome syntenic blocks for each subgenome and inferring ancestral genomes.

We first used Orthofinder[64] to detect orthogroups based on protein sequences of four *Papaver* species and *Corydalis tomentella*[80] and transformed the genomes into orthogroup ID sequences (Orthseq). Then, we used Orthseq of four *Papaver* species to build syntenic blocks. We filtered out orthogroups with gene copy numbers larger than their target copy number (e.g., two for WGD and one for no WGD) in at least one species. We then used DRIMM-Synteny[73] with default parameter to build initial syntenic blocks (v1-SynBs). For each v1-SynB, we identified the longest common subsequence (LCS) between the block Orthseq and initial Orthseq to obtain the start and end genes, from which we determined the genomic coordinates of each v1-SynB. We discarded those v1-SynBs whose copy number was unequal to the target copy number in each species, and then detected the subgenome v1-SynBs based on the overlapping of subgenomes coordinates with v1-SynBs coordinates. We split each chromosome's v1-SynBs into subgenome-specific v1-SynBs. We assigned subgenome ID for the orthogroups of the corresponding subgenome-specific v1-SynBs. Then we re-ran DRIMM-Synteny with parameter cycle length 60 to obtain new syntenic blocks (v2-SynBs), and then applied the LCS algorithm to derive genomic coordinates of v2-SynBs in each species (Table S14).

On review, we found that a syntenic block was missed by DRIMM-Synteny (identified as block B1 in Figure S23 and Table S15). We manually added the B1 block into each species and obtained final syntenic blocks (final-SynB) for each species (Figure S23 and Table S15). Based on the subgenome ID of each orthogroup, we obtained final-SynBs for each subgenome. We then inferred final-SynB connections for each chromosome of each subgenome based on connection count and centromere location. Specifically, for each subgenome's final-SynB, we recorded its endpoint connections in other subgenomes and species and inferred its connecting partner as the final-SynB with the largest connection count which does not break existing connections. After this initial round of connection, we manually clarified any ambiguous connections by manual review of the final-SynB connections in *P. rhoeas* and *P. bracteatum,* under the assumption that each chromosome would have at least one centromere (Table S15).

Based on the subgenome/species chromosome final-SynBs and the subgenome-aware phylogenetic tree, we inferred five ancestral genomes (Table S15). Ancestors 1 to 4 were inferred by IAGS GMP model.[32] The inference of ancestor 1 used the block sequences of subgenomes 1, 4 and *P. rhoeas,* ancestor 2 used subgenome 2, ancestor 1 and *P. rhoeas*, ancestor 3 used subgenome 3, *P. rhoeas* and *P. bracteatum*, and ancestor 4 used ancestor 2, ancestor 3 and *P. bracteatum*.

We used IAGS to count shuffling events across the phylogenetic tree and reconstruct the evolutionary history of *Papaver* (Figure 4; Table S15).

## Reconstruction of cenSat evolution trajectory

The evolutionary trajectories of four *Papaver* cenSats, PCEN238, αPCEN168, PCEN338 and αPCEN169, were inferred based on the subgenome-aware phylogenetic tree and parsimonious assumption. First, for each cenSat, we queried it in every genome using Lastz[70] retaining only those hits with identity >80% and alignment length >80% of query cenSat length (Figure 3 and Table S12). After querying, we found four cenSats coexisted in all four species with different copy numbers (Figure 3A). The cenSat amplified in most chromosomes of one species or one subgenome was defined as a "potential functional satellite", like PCEN238 in *P. bracteatum* (amplified in 4 chromosomes with 31,854 copies), αPCEN168 in *P. rhoeas* (amplified in all 7 chromosomes with 188,050 copies) and SG4 (35,615 copies), PCEN338 in SG1 and SG2 (71,164 copies), and αPCEN169 in SG3 (24,625 copies) (Table S12).

As described above, we reconstructed the karyotypes of five ancestors in the subgenome-aware phylogenomic tree through IAGS framework.[32] Specifically, ancestor 5 is the most recent common ancestor (MRCA) of the four *Papaver* species, ancestor 4 is the MRCA of *P. setigerum*, *P. somniferum* and *P. rhoeas*, ancestor 3 is the MRCA of subgenome S3 and *P. rhoeas*, ancestor 2 is the MRCA of three subgenomes SG1, SG4 and SG2, and ancestor 1 is the MRCA of tow subgenomes SG1 and SG4 (Figure 4).

We inferred the potential functional cenSats in each ancestor (Figure 4). For ancestor 5 at 8.2 Ma, PCEN238 was amplified in 4 chromosomes with 31,854 copies of *P. bracteatum* and we also found it was amplified in chr4 and 8 of *P. somniferum* with 2,892 and 3,419 copies respectively, amplified in chr1, 8, and 15 of *P. setigerum* with 1,727, 2,716, and 1,103 copies, respectively. Therefore, we inferred that PCEN238 was the potential functional cenSat of ancestor 5.

For ancestor 4 at 7.4Ma, αPCEN168 was amplified in all 7 chromosomes of *P. rhoeas* with 188,050 copies, and we also found it was amplified in all 7 chromosomes of SG4 with 35,615 copies. Therefore, we inferred that αPCEN168 was the potential functional cenSat of ancestor 4.

Based on ancestor 4 and *P. rhoeas*, we inferred that potential functional cenSat of ancestor 3 was αPCEN168 and was replaced by αPCEN169 in SG3.

Ancestor 2 was more complex, since PCEN338 amplified in both SG1 and SG2, however, SG4 was the closest subgenome to SG1 on the phylogenetic tree. Here, we proposed three hypotheses: the first hypothesis: PCEN338 was amplified in ancestor 2 at 6.1 Ma

and it was the potential functional satellite of ancestors 1 and 2. Subsequently, SG4 lost its PCEN338 sequences and re-amplified αPCEN168; the second hypothesis: αPCEN168 was the potential functional satellite of ancestors 1 and 2. Then, PCEN338 independently amplified in both SG1 and SG2; the third hypothesis: αPCEN168 was the potential functional satellite of ancestors 1, 2, SG1, SG2 and SG4. Then, PCEN338 amplified after the hybridization of SG1 and SG2.

Compared to the first and second hypotheses, the third one is the most parsimonious, as it does not postulate either deletion and re-amplification (first hypothesis) or independent amplification (second hypothesis). Therefore, we assumed the third hypothesis represented the most probable order of amplification for the centromere satellites. However, we could not rule out alternative explanations.

**QUANTIFICATION AND STATISTICAL ANALYSIS**

The statistical methods used in this study are indicated in the figures, figure legends, and methods. Statistical analyses were performed using R v4.2.2 and the ggplot2 v3.4.0 package.

