## [Document S2. Transparent peer review records for Shenghan Gao et al · Cell Genomics]

The centromere landscapes of four karyotypically diverse *Papaver* species provide insights into chromosome evolution and speciation

Shenghan Gao (高胜寒), Yanyan Jia (贾彦彦), Hongtao Guo (郭弘涛), Tun Xu (徐瞰),
Bo Wang (王博), Stephen J. Bush, Shijie Wan (万世杰), Yimeng Zhang (张一蒙), Xiaofei
Yang (杨晓飞), Kai Ye (叶凯)

Summary

Initial submission: Received : Dec 18, 2023

Scientific editor: Sara Rohban

First round of review: Number of reviewers: 3
Revision invited : Feb 07, 2024
Revision received : Apr 16, 2024

Second round of review: Number of reviewers: 3
Revision invited : May 14, 2024
Accepted : Jul 09, 2024

Data freely available: YES

Code freely available: YES

This transparent peer review record is not systematically proofread, type-set, or edited. Special characters, formatting, and equations may fail to render properly. Standard procedural text within the editor's letters has been deleted for the sake of brevity, but all official correspondence specific to the manuscript has been preserved.

Referees' reports, first round of review

Reviewer #1:

This study performs long read sequencing and genome assembly in four *Papaver* species and derives insights into chromosome and centromere evolution. Two species were diploid, and two were polyploid, providing insight into how centromere evolution occurs in polyploids.

Assembly was made using ~47x HiFi reads, and Hi-C short reads for scaffolding and appropriate validation performed. Were the plants sequenced heterozygous at all?

Gene and transposon annotation are very briefly discussed. I would be interested to see a more developed discussion of transposon annotation, and whether particular classes or families have expanded or contracted across these species?

Importantly, the authors also validate centromere location with CENH3 CHIP-seq - this is not mentioned in detail in the main text and should be expanded upon.

Like many plants, the *Papaver* genomes have satellite repeat centromeres. The authors use TRF to map the tandem repeats - other methods are available for de novo mapping of TRs, including TRASH and SRF, that may be worth trying and comparing to.

Detailed analysis is performed on the satellite repeat families identified, and FISH performed in some cases to validate their location at centromeres. Higher order repeat analysis is performed. On line 154 it would be helpful if the authors could describe their method for HOR mapping - I also couldn't find this information in the methods.

To what extent were transposons found within the centromere satellite repeats - analogous to ATHILA insertions in *Arabidopsis thaliana*? In Figure 1 it seems like there are abundant LTR elements in the CENH3 occupied regions - what are these elements? Eg Are they CRM, ATHILA, Copia etc?

Some of the patterns in Figure 1 are intriguing in potentially showing two discrete CENH3 peaks - these should be commented on in more detail in the main text - this could show two discrete centromere regions, or possibly single centromeres in different cells that are merged in this population analysis.

Figure S9 would be good to include as one of the main figures. Same comment for S19, although more detail on how this was generated needs to be provided.

Interestingly, the dynamics of satellites in the polyploid subgenomes is complex. This might be interesting to compare to the situation in *Arabidopsis suecica* allopolyploids, where the individual subgenome satellites have remained isolated. Comparison of karyotype evolution implies a role for centromere dynamics in rearrangements.

The centromere drive model discussed is interesting, but how the drive would occur should be explained in more detail - eg female meiotic drive, or another effect? With respect to this model, further comparison of CENH3 amino acid sequence amongst the *Papaver* species, and perhaps related plants, would be interesting to see. Eg One might expect higher levels of amino acid diversity in CENH3

compared to other genes, according to this model? This is shown in Figure S15 but not really discussed in detail in the main text.

Minor points

Line 44 - not all centromeres are composed of satellite arrays, eg budding yeast.

Line 155 - some centromeres are AT rich, but not all. For example the Arabidopsis thaliana centromeres are relatively GC rich.

Reviewer #2:

In this manuscript, the authors generated nearly complete genome assemblies for four karyotypically diverse Papaver species. They identified centromere satellite families with experimental validation. They characterized the distribution and analyzed the evolution of satellites during Papaver hybridization. This study presented comprehensive analysis of Papaver centromere and satellites, provided insight into centromere evolution of hybridization and speciation. However, some issues can be further addressed.

1. Why is the assembly quality of *P. rhoeas* much worse than the other three, with QV of only 22? And there is also a huge difference between it and the previous assembly version. An improved *P. rhoeas* assembly is required.
2. The numbers of *P. rhoeas* "assembly after 3d-DNA" in Table S2 seem incorrect. In addition, I am not sure if using 3d-DNA as a reference and then using ragtag-scaffold could improve assembly quality. Why not use the latest version of hifiasm or Verkko to assemble HiFi and ONT data together to improve the contig quality, instead of assembling contigs, separately?
3. Figure S16 PsoCENH3 CHIP-seq has multiple peaks on the same chromosome with a large span. How can we determine the exact location of the centromere? And the centromere region identified in Fig. 1D of the article is much larger than that of other published genomes.
4. Pso338S1 appears in the centromeric region of *P. somniferum*, and in regions outside the centromere, but not in the centromeres of some chromosomes in *P. setigerum* (Figure 2D), please explain it.

Reviewer #3:

Centromeres are important part of genome for essential biological function, while it is difficult to assemble as the complex structure. This paper carried out good research of the centromeres evolution of Papaver species, providing a comprehensive satellite library. Some minor issues as follow could be refined further before it is published.

Introduction section: The background information is not sufficient, and relevant information needs to be added, such as additional filament particles and their research progress, horizontal comparison, etc.

The community detection method was initially employed to construct 124 satellites in the range of line

112-118, followed by screening 26 candidate cenSats using the criterion of 'excluding satellites with a genomic size < 1 Mb or AT content < 60%'. What are the reasons behind these treatment criteria?

The pictures and tables need to be improved. For example, Table 1: label the data units could be consistently in the first column; Table S11: as this big table contained multiple information of total four species, I think information of cenSat of different species could displayed in different sheet or any other effective formation for improving the data usability.

In line 161, can you clarify how the 'suggesting in both cases that they were recent additions to the genome' is concluded?

Modified the language formatting in both the main text and annexes to make it more consistent, such as removing multiple definitions of the same abbreviation, using English numbers consistently, correcting improper use of units (e.g., changing 'x' to appropriate unit for depth), and ensuring uniformity of font sizes.

6. Are there limitations to the application of the cross-comparison Model to other species?

Authors' response to the first round of review

We thank the reviewers for their constructive comments and provide individual responses to each of them below. We hope that this revised version of our manuscript is suitable for publication in *Cell Genomics*.

Reviewer #1:

This study performs long read sequencing and genome assembly in four Papaver species and derives insights into chromosome and centromere evolution. Two species were diploid, and two were polyploid, providing insight into how centromere evolution occurs in polyploids.

Assembly was made using ~47x HiFi reads, and Hi-C short reads for scaffolding and appropriate validation performed. Were the plants sequenced heterozygous at all?

RESPONSE: For *P. bracteatum*, we generated the HiFi and Hi-C data from the same plant, but for *P. rhoeas*, *P. somniferum*, and *P. setigerum*, we integrated newly sequenced HiFi data with data we previously published in Yang X. et al. *Nature Commu.*, 2021. Although the newly sequenced HiFi data and previously published data were from different plants, both plants were sprouted from seeds of the same capsule.

Gene and transposon annotation are very briefly discussed. I would be interested to see a more developed discussion of transposon annotation, and whether particular classes or families have expanded or contracted across these species?

RESPONSE: We have expanded the description of gene and repetitive element annotation in the ‘genome annotation’ section of the Method Details part at lines 684 to 738, and expanded the discussion of these results (at lines 103 to 112) “Approximately three quarters of each of the four *Papaver* genomes comprised repetitive elements, with long terminal repeat (LTR) retrotransposons the predominant repeat family, comprising on average 54% of the genome (Tables 1 and S4). Each of the four genomes was broadly consistent in its composition of LTR retrotransposons, with Gypsy elements accounting for 22.8-27.5% of the genome, and Copia elements 15.6-25.1% (Table S4). However, we observed that *P. bracteatum* was at the extremes of each range, having a genome comprising 15.6% Copia and 27.5% Gypsy elements, suggesting a relative contraction of the former and expansion of the latter (Table S4). In addition, we found that, on average, only 4.5%, 3.7%, and 0.04% of the genome could be annotated as DNA transposons, LINES and SINES, respectively (Table S4)”.

Importantly, the authors also validate centromere location with CENH3 CHIP-seq - this is not mentioned in detail in the main text and should be expanded upon.

RESPONSE: We agree that it would be clearer to elaborate on this aspect of the methodology, and have added a ‘ChIP-seq data analysis’ section to the methods (lines 864 to 876) and revised the main text (from lines 148 to 159) beginning “To complement this finding, we also performed chromatin immunoprecipitation followed by sequencing (ChIP-seq) with the PsoCENH3 antibody (centromeric histone H3, encoded by *P. somniferum* gene *Pso04G02820.1*) (Figure S14) and aligned the data to the assembled genome, finding significantly higher read coverage at the Pso338S1 locus in each of the 11 chromosomes, and supporting our interpretation of Pso338S1 as a prevalent cenSat in *P. somniferum* (Figures 2D and S15). Specifically, we observed, on average, 4-fold ChIP/input enrichment within the identified Pso338S1 array regions, compared with the other genomic regions (Table S9). A number of centromeres, including chromosomes 9, 10 and 11, also showed multiple discrete PsoCENH3 peaks (Figure 2D), suggesting that they either contained multiple discrete centromere regions or, alternatively, that individual centromeres from different cells were merged in our population-scale analysis”.

*Like many plants, the *Papaver* genomes have satellite repeat centromeres. The authors use TRF to map the tandem repeats - other methods are available for de novo mapping of TRs, including TRASH and SRF, that may be worth trying and comparing to.*

RESPONSE: We agree that other methods would also be worth exploring, and so re-performed the tandem repeat detection step using both TRASH and SRF. The TRASH output files indicated that (as with TRF) each tandem repeat region could be isolated. Consequently, we calculated the pairwise edit distance between the *consensus.primary* sequences from the TRASH *summary.csv* and subsequently clustered them. Compared to our TRF-based method, TRASH yielded consistent results for the four species, illustrated below (**Figure R1A-D**), and as it does not meaningfully alter our conclusions, we retain data from TRF in the main text. By contrast, SRF, as described on their GitHub (<https://github.com/lh3/srf>), is designed primarily for the identification of higher-order repeats (HORs) themselves rather than the minimal repeat unit within a given HOR. This makes it challenging to directly compare SRF results with those from either TRASH or our TRF-based method. For instance, in the Prh168S1 region of *P. rhoeas* chromosome 1 (chr1:119,772,951-166,516,112), the top two repeats

identified by SRF were HOR circ10-503 (comprising three monomers) and circ25-1005 (comprising six monomers), rather than the minimal repeat unit (**Figure R1E-F**).

Figure R1. Tandem repeat mapping based on TRASH and SRF. A-D. Centromere satellites mapping with TRASH in *P. bracteatum* (A), *P. rhoeas* (B), *P. somniferum* (C) and *P. setigerum* (D). **E.** SRF annotation repeat frequencies for the *P. rhoeas* chromosome 1 centromere satellite array region. The top two items are the higher order repeats circ10-503 (with unit length 503bp) and circ25-1005 (with unit length 1005bp). **F.** Dotplots between Prh168S1 and circ10-503, as well as circ25-1005.

Detailed analysis is performed on the satellite repeat families identified, and FISH performed in some cases to validate their location at centromeres. Higher order repeat analysis is performed. On line 154 it would be helpful if the authors could describe their method for HOR mapping - I also couldn't find this information in the methods.

RESPONSE: We have added a new section to the methods, “Higher-order repeat (HOR) annotation”, from lines 817 to 829.

To what extent were transposons found within the centromere satellite repeats - analogous to ATHILA insertions in Arabidopsis thaliana? In Figure 1 it seems like there are abundant LTR elements in the CENH3 occupied regions - what are these elements? Eg Are they CRM, ATHILA, Copia etc?

RESPONSE: This is a good observation – we agree that there do appear to be abundant LTR elements in the CENH3-occupied regions (as throughout the rest of the genome – which we have discussed further above). Accordingly, we have revised Figure 1D (now Figure 2D in the updated manuscript, and shown as **Figure R2** below) to now include LTR subfamily tracks (LTR-Gypsy, LTR-Copia, LTR-other and LINE) for the centromeric regions. In addition, we did not find ATHILA and CRM in the PsoCENH3 occupied regions.

Figure R2. Sequence coverage of PsoCENH3 ChIP-seq data on each inferred (peri-) centromere region in *P. somniferum*. Tracks from bottom to top indicate the location of genes, transposable elements (LTR-Copia, LTR-Gypsy, Other LTR, and LINES), and Pso338S1 satellite arrays, respectively.

Some of the patterns in Figure 1 are intriguing in potentially showing two discrete CENH3 peaks - these should be commented on in more detail in the main text - this could show two discrete centromere regions, or possibly single centromeres in different cells that are merged in this population analysis.

RESPONSE: We certainly agree that this is an intriguing observation, and have now made reference to the plausible explanations for it in lines 155 to 159; this is quoted above in response to an earlier comment. Ultimately, we think that single-cell long-read DNA sequencing technology will be instrumental in further investigating this phenomenon, but that this is beyond the scope of the present work.

Figure S9 would be good to include as one of the main figures. Same comment for S19, although more detail on how this was generated needs to be provided.

RESPONSE: We have moved Figure S9 to the main manuscript, presenting it now as Figure 1; we have also moved Figure S19 to the new Figure 3F.

*Interestingly, the dynamics of satellites in the polyploid subgenomes is complex. This might be interesting to compare to the situation in *Arabidopsis suecica* allopolyploids, where the individual subgenome satellites have remained isolated. Comparison of karyotype evolution implies a role for centromere dynamics in rearrangements.*

RESPONSE: We agree that there is likely a (complex) role for centromere dynamics in genomic rearrangement, although to fully unpack the concept would necessitate additional data and not be practicable in the present work. Nevertheless, the point of contrast with *Arabidopsis suecica* allopolyploids (Burns, et al. *Nat Ecol Evol*, 2021, PMID: 34413506) is an important one, and to that end we now refer to it on lines 254 to 260: "We observed six types of chromosomal rearrangement, affecting 15 chromosomes, nine of which involved centromeric sequence (Figures 5A and S22). These observations support a possible relationship between the centromere dynamics of the subgenomes and structural rearrangements within them, one more complex than (for instance) that observed in *Arabidopsis suecica*, in which there was little evidence of 'genomic shock' following its polyploidization, with no subgenome dominance in expression, seemingly isolated cenSats, and no apparent genomic reorganization"

The centromere drive model discussed is interesting, but how the drive would occur should be explained in more detail - eg female meiotic drive, or another effect?

RESPONSE: We have substantially revised the introduction at lines 53 to 65 (as suggested by Reviewer #3 Comment #1) to elaborate on the mechanism of centromere drive; with regard to our model, our interpretation is that it is driven by asymmetric female meiosis.

With respect to this model, further comparison of CENH3 amino acid sequence amongst the *Papaver* species, and perhaps related plants, would be interesting to see. Eg One might expect higher levels of amino acid diversity in CENH3 compared to other genes, according to this model? This is shown in Figure S15 but not really discussed in detail in the main text.

RESPONSE: To investigate the diversity of the CENH3 amino acid sequence, we first identified the CENH3 gene as *Pbr03G53170.1* in *P. bracteatum* by leveraging its syntenic relationship with *PsoCENH3*. We then determined the amino acid sequence identity for syntenic gene pairs between *P. bracteatum* and *P. somniferum*, as well as between *P. bracteatum* and *P. setigerum* by BlastP, highlighting the *PsoCENH3*-related gene pairs in **Figure R3**, below. Our analysis revealed that the sequence identity of *PsoCENH3*-related genes is situated to the left of the peaks, suggesting a higher level of amino acid diversity in CENH3 compared to other genes. We describe this finding on lines 338 to 345 and have added **Figure R3** to the manuscript as Figure S28C.

Additionally, we compared the amino acid sequence between *Papaver* species and *Arabidopsis* (HTR12, UniProt ID: Q8RVQ9) using CLUSTAL O (version 1.2.4, <https://www.ebi.ac.uk/jdispatcher/msa/clustalo>), as shown in **Figure R4**. We observed that the Nterminal tail of *PsoCENH3* and its syntenic pairs exhibit diversity, whereas the histone fold domains are relatively conserved. The demarcation between the N-terminal tail and the histone fold domain is based on the findings from Talbert, et al, *Plant Cell*, 2002 (PMID: 12034896).

Figure R3. The amino acid sequence identity for syntenic gene pairs between *P. bracteatum* and *P. somniferum*, as well as between *P. bracteatum* and *P. setigerum*, highlighting the *PsoCENH3*-related gene pairs and the median value.

CLUSTAL O(1.2.4) multiple sequence alignment

Figure R4. Multiple sequence alignment of the amino acid sequences of *PsoCENH3* and its syntenic genes in *Papaver* species as well as *HTR12*, the *CENH3* gene of *Arabidopsis*.

Minor points

Line 44 - not all centromeres are composed of satellite arrays, eg budding yeast.

RESPONSE: We have revised this sentence to clarify that “Centromeres of many species comprise large arrays of tandemly repeated satellite DNA often dominated by one repeat” (in lines 44 to 45).

Line 155 - some centromeres are AT rich, but not all. For example the Arabidopsis thaliana centromeres are relatively GC rich.

RESPONSE: We have revised the sentence to read “as centromeric sequence is often characteristically AT-rich” (in line 132).

Reviewer #2: Comments enter in this field will be shown in this manuscript, the authors generated nearly complete genome assemblies for four karyotypically diverse Papaver species. They identified centromere satellite families with experimental validation. They characterized the distribution and analyzed the evolution of satellites during Papaver hybridization. This study presented comprehensive analysis of Papaver centromere and satellites, provided insight into centromere evolution of hybridization and speciation. However, some issues can be further addressed.

1. Why is the assembly quality of P. rhoeas much worse than the other three, with QV of only 22? And there is also a huge difference between it and the previous assembly version. An improved P. rhoeas assembly is required.

RESPONSE: We agree that *P. rhoeas* is of poorer quality than the other three genomes, but do not believe that this has a material impact on our conclusions, or that an improved assembly is realistically possible with our current data (which already comprises HiFi, ONT and Hi-C reads). It is also important to note that *P. rhoeas* can be distinguished from the other three species in being self-incompatible, which would result in a higher heterozygosity (Daphne et al, *Curr Biol.*, 2023, PMID: 37279687) (the other three *Papaver* species are self-compatible). We have evaluated the heterozygosity of *Papaver* species in our previous published work, and the heterozygosity of *P. rhoeas* is as high as 3.18% (Yang et al. *Nature Communications*, 2021), far higher than, for example, 0.7% for *P. bracteatum* (Figure S3). Another factor which may explain the lower QV is the source material used for *P. rhoeas*. We obtained the NGS data from our previous study (Yang, et al., 2021) and for the current work sequenced HiFi reads, which we used for assembly; as such, the material was obtained from two plants. Although these two plants were sprouted from seeds of the same capsule, they still differ more substantially in their heterozygosity than either *P. somniferum* or *P. setigerum* as a consequence of the self-incompatible nature of *P. rhoeas*, resulting in a comparatively lower QV. We have added an explanatory note in Table S2 to explain this.

Nevertheless, to confirm the quality of the *P. rhoeas* assembly, we re-evaluated the difference between our assembly and the previous assembly chromosome by chromosome (**Figure R5**) using Hi-C data. We have a number of substantial improvements of the previous assembly, including:

- The contig N50 is improved from 5.29 Mb (previous genome) to 29.07 Mb (current genome), a sixfold increase;
- In our assembly, we successfully assembled the centromere sequences, which are fragmented satellites in the previous one;
- We corrected the ubiquitous large inversions of the previous assembly, shown as green boxes in **Figure R5**;
- We removed the large duplicated sequences of the previous assembly, shown as blue boxes in **Figure R5**.

Taken together, these observations indicate that the current *P. rhoeas* genome is substantially improved compared to the previously published one.

Figure R5. Dotplot and Hi-C heatmap comparison of our *P. rhoeas* assembly and that published in 2021.

2. The numbers of *P. rhoeas* "assembly after 3d-DNA" in Table S2 seem incorrect. In addition, I am not sure if using 3d-DNA as a reference and then using ragtag-scaffold could improve assembly quality. Why not use the latest version of hifiasm or Verkko to assemble HiFi and ONT data together to improve the contig quality, instead of assembling contigs, separately?

RESPONSE: Table S2 shows increases in the number of contigs between the 'assembly by hifiasm' and 'assembly after 3d-DNA' columns for each species, although we note that they are highly variable. For instance, there is a minimal increase from 2768 to 2826 contigs for *P. setigerum* but an approx. five-fold increase from 1829 to 10,408 for *P. somniferum*. For *P. rhoeas*, the increase is far more substantial, from 371 to 25,968 contigs. We agree that these numbers may seem unusual at first glance but after double-checking we do believe they are correct. As to what underlies the disparity, in general, the 3d-DNA method constructs chromosome-scale scaffolds but may fragment large contigs into smaller pieces due to false positive mis-join signals, likely arising from relatively lower-quality Hi-C data aligning to larger, more repetitive, contigs. This fragmentation would reduce the contig N50 and increase the number of contigs in the overall assembly. A similar phenomenon has been reported by Wang *et al.* (PMID: 36482318) who note that "3D-DNA cuts contigs in the middle site to form sister-contigs" leading to an increase in the contig number of *P. rhoeas* after 3d-DNA scaffolding (as discussed above, we observe an increase in contig number for each of the other assemblies too, albeit to a lesser degree). We adopted our current assembly strategy in light of these findings by Wang, *et al.* Specifically, we just utilized 3d-DNA for the construction of the chromosome structures, which serve as a reference to guide us for scaffolding the contigs into chromosomes using ragtag-scaffold. The contigs in our assembly are from hifiasm (v0.16.1) and the ragtag-scaffold step does not change any contig sequence. Overall, we attribute to this strategy the high scaffold N50s reported in this work, although we are mindful that ongoing improvements to assembly algorithms may also be of benefit.

Further to your suggestion, we used an updated version of hifiasm 0.19.6 (released August 2023) to re-assemble HiFi and ONT data together for *P. rhoeas* genome, and obtained a contig N50 of 29.79 Mb, only marginally higher than that already presented (29.07 Mb). This minimal improvement will not change the conclusion of our work.

3. Figure S16 PsoCENH3 ChIP-seq has multiple peaks on the same chromosome with a large span. How can we determine the exact location of the centromere? And the centromere region identified in Fig. 1D of the article is much larger than that of other published genomes.

RESPONSE: We agree that determining the exact location of the centromere is challenging, and we address the issue of multiple peaks in response to reviewer 1, above (comment #7). With regard to Figure 2D (Figure 1D in the initial submitted version), the figure shows both the centromere and pericentromere, so would seem larger than that of other genomes. We provide the inferred location of centromere satellite arrays in Table S9.

4. *Pso338S1* appears in the centromeric region of *P. somniferum*, and in regions outside the centromere, but not in the centromeres of some chromosomes in *P. setigerum* (Figure 2D), please explain it.

RESPONSE: Based on the satellite library hypothesis, related species share a satellite DNA library, and one species will select one satellite to amplify forming the centromere satellite arrays (Ugarkovic, *The Open Evolution Journal*, 2008; Fry et al. *Cell*, 1977). In our work, we reconstructed the evolutionary history of *Papaver* centromere satellites based a subgenome-aware analysis (Figure 4), and showed a consistent result with satellite library hypothesis. That is α PCEN338 (*Pso338S1* in *P. somniferum*) was selected to amplify after the hybridization of SG1 and SG4 forming pre-*P. somniferum* while α PCEN168 and α PCEN169 were selected to amplify after the hybridization of SG2 and SG3 forming pre-*P. setigerum*-1. Therefore, α PCEN338 (*Pso338S1*) were not amplified in the chromosomes from pre-*P. setigerum*-1 (including chr6, chr7, chr11, chr13, chr16, chr17, chr18, chr19, chr20, chr21, and chr22) in *P. setigerum* and showed the patterns in Figure 3D (Figure 2D in initial submitted version).

Reviewer #3: *Centromeres are important part of genome for essential biological function, while it is difficult to assemble as the complex structure. This paper carried out good research of the centromeres evolution of Papaver species, providing a comprehensive satellite library. Some minor issues as follow could be refined further before it is published.*

Introduction section: The background information is not sufficient, and relevant information needs to be added, such as additional filament particles and their research progress, horizontal comparison, etc.

RESPONSE: We have expanded the introduction to provide additional background to the work and in conjunction with other reviewer comments regarding presentation and clarification of other issues, hope this revised version provides a more comprehensive overview.

The community detection method was initially employed to construct 124 satellites in the range of line 112-118, followed by screening 26 candidate cenSats using the criterion of 'excluding satellites with a genomic size < 1 Mb or AT content < 60%'. What are the reasons behind these treatment criteria?

RESPONSE: Based on the review of Talbert and Henikoff (Paul B Talbert, et al, *Exp Cell Res*, 2020; PMID: 32035948), in most plants and animals, satellite centromere sequences tend to be AT-rich and megabase-length, therefore we used initial criteria of < 1 Mb and < 60% to narrow down and select appropriate candidate satellites. In addition, it is important to note that the final set of centromere satellites were eventually determined by syntenic relationships and experiments.

The pictures and tables need to be improved. For example, Table 1: label the data units could be consistently in the first column; Table S11: as this big table contained multiple information of total four species, I think information of cenSat of different species could displayed in different sheet or any other effective formation for improving the data usability.

RESPONSE: Thanks for your suggestion. We have revised the figures and tables based on your suggestions.

In line 161, can you clarify how the 'suggesting in both cases that they were recent additions to the genome' is concluded?

RESPONSE: Based on the model of satellite centromere life cycle reviewed by Talbert and Henikoff (Talbert, et al, *Exp Cell Res*, 2020; PMID: 32035948), a breakage event can lead to the loss of an 'old' centromere with neo-centromeres often forming 'near' the previous one. Consequently, we consider that centromere satellites which are detected in multiple chromosomes are more likely to be ancestral ones and that those which are primarily found only on one chromosome are more likely to be recent additions to the genome, *i.e.* neocentromere satellites.

We revised the sentence to read " The isolated node β PCE168 was only detected on *P. setigerum* chr21 (with 15,941 copies), whilst β PCE169 expanded primarily on *P. bracteatum* chr1 with 11,370 copies (Figure 3B, D; Table S11), and suggesting in both cases that they may be neocentromere satellites (*i.e.* 'neo-arrays')" (lines 187 to 190).

Modified the language formatting in both the main text and annexes to make it more consistent, such as removing multiple definitions of the same abbreviation, using English numbers consistently, correcting improper use of units (e.g., changing 'x' to appropriate unit for depth), and ensuring uniformity of font sizes.

RESPONSE: Thanks for your comments. We have revised the language and format throughout, and hope the revised version is consistent and clear. For example, we revised 'x' to '×', which is the widely used unit for depth.

6. Are there limitations to the application of the cross-comparison Model to other species?

RESPONSE: Our cross-comparison model is designed for the complex evolutionary history of the *Papaver* genus, which comprises subgenome evolution, (competing) satellite-type centromeres, and two rounds allopolyploidization, and its associated genome rearrangements. We believe that our model would apply to species from the same genus and have sufficient synteny but given the complexity of this evolutionary scenario, would approach other species with caution. In particular, we think that our model would not be readily applicable to other types of centromeres, such as point centromere (*e.g.* budding yeast) or transposon-based centromeres (*e.g.* in green algae).

Referees' report, second round of review

Reviewer #1: Thank you - the authors have attended to my comments fully.

Reviewer #2: Having reviewed the revisions and the detailed responses provided by the authors, I am satisfied with most of the manuscript. However, I will keep my concern about the low assembly quality of *P. rhoeas*. The QV of 22 indicates low accuracy and completeness, especially for highly repetitive regions. Thus, I am still worried about the completeness of centromere identification of *P. rhoeas*. Is the low-quality assembly of *P. rhoeas* related to only 100.7 Mb-length tandem repeats detecting in *P. rhoeas*, and noticeably, only one cenSat detected in *P. rhoeas*? And for the high heterozygosity as 3.18% of *P. rhoeas*, I think the haplotype-resolved assembly is required.

Reviewer #3: The authors have made detailed revisions to the manuscript in response to the issues. I have no other questions and agree to accept this work.

Authors' response to the second round of review

We thank the reviewers for their constructive comments and provide individual responses to each of them below. We hope that this revised version of our manuscript is suitable for publication in *Cell Genomics*.

Reviewers' Comments:

Reviewer #1: Thank you - the authors have attended to my comments fully.

RESPONSE: Thanks for your valuable comments for improving our manuscript.

*Reviewer #2: Having reviewed the revisions and the detailed responses provided by the authors, I am satisfied with most of the manuscript. However, I will keep my concern about the low assembly quality of *P. rhoeas*. The QV of 22 indicates low accuracy and completeness, especially for highly repetitive regions. Thus, I am still worried about the completeness of centromere identification of *P. rhoeas*. Is the low-quality assembly of *P. rhoeas* related to only 100.7 Mb-length tandem repeats detecting in *P. rhoeas*, and noticeably, only one cenSat detected in *P. rhoeas*? And for the high heterozygosity as 3.18% of *P. rhoeas*, I think the haplotype-resolved assembly is required.*

RESPONSE: Thanks for your valuable question. To address concerns regarding the assembly quality of *P. rhoeas* and the accuracy and completeness of centromere satellite identification in *P. rhoeas*, we conducted a thorough reevaluation. This involved assessing the QV using HiFi reads from the same individual plant, performing CHIP-seq analysis of PrhCENH3, and generating a draft haplotype-resolved genome for *P. rhoeas*. All of these evidences strongly support both the highquality assembly of *P. rhoeas* and the accuracy and completeness of our centromere satellite identification in *P. rhoeas*. Details as following:

- We reassessed the QV of the *P. rhoeas* genome using high-fidelity (HiFi) reads, verifying the high-quality of our assembly. We achieved a QV of 67.79 by Merqury software (version 1.3) based on the HiFi reads from the same plant individual, suggesting the high-quality of assembly. Due to high heterozygosity (3.18%) and self-incompatible nature of *P. rhoeas*, we obtained the lower value, *e.g.* QV of 22, when we evaluated the QV with short-reads sequencing data from a different plant individual.
- We performed additional CHIP-seq of CENH3 in *P. rhoeas*, revealing the accuracy of the centromere satellite identification. Synteny analysis between *P. rhoeas* and *P. somniferum* revealed the gene coding CENH3 in *P. rhoeas* as *Prh03G45160.1*, which was further supported by protein sequence alignments between *Prh03G45160.1* and the CENH3 in other three species (*P. somniferum*: PsoCENH3, *Nelumbo nucifera*: NnCenH3-A, and *Arabidopsis thaliana*: HTR12_ARATH) (**Figure R1**). We named *Prh03G45160.1* as *PrhCENH3*. We performed CHIP-seq for PrhCENH3 antibody and aligned the reads to the assembled genome. This new data confirmed a match with the identified centromere satellite array of Prh168S1 (**Figure R2**), thereby validating the accuracy of the centromere satellite.
- We generated the draft haplotype-resolved genome of *P. rhoeas* and identified the corresponding centromere satellite, confirming the completeness of centromere satellite identification in our current study. We reassembled the genome using hifiasm (v0.19.6) to obtain the draft haplotype-resolved genome of *P. rhoeas* (H1 and H2) and identified the corresponding centromere satellite as Prh168S2_H1 and Prh168S1_H2, respectively. We compared the sequence similarity (**Figure R3**) and revealed that Prh168S2_H1, Prh168S1_H2 belong to the same satellite type of Prh168S1 (the current identified centromere satellite in *P. rhoeas*). These findings imply that the haplotype-resolved genome does not leads to new centromere satellite in *P. rhoeas*, therefore, we used collapsed genome assembly of *P. rhoeas* in our study, consisting with other species.

We have submitted the CHIP-seq data of PrhCENH3 to the NGDC under accession number of CRA016610 (<https://ngdc.cncb.ac.cn/gsa/browse/CRA016610>). We have made the corresponding revisions in the main manuscript at lines from 138 to 144 and in the Methods section at lines from 854 to 869. Additionally, the draft haplotype-resolved assembly of *P. rhoeas* has been uploaded to the NGDC with the accession numbers GWHESSC00000000 (<https://ngdc.cncb.ac.cn/gwh/Assembly/84865/show>) and GWHESSB00000000 (<https://ngdc.cncb.ac.cn/gwh/Assembly/84863/show>). Figures R1 and R2 have been included as Supplementary Figures S12 and S13.

A CLUSTAL O(1.2.4) multiple sequence alignment

 CENH3

B

	Pso04G02820.1	NnCenH3-A	HTR12_ARATH
Prh03G45160.1	83.45% (top 1 hit)	66% (top 1 hit)	54.95% (top 1 hit)

Figure R1. Identification of *PrhCENH3* gene as *Prh03G45160.1*.

(A) Cluster Omega alignment of protein sequences of *Prh03G45160.1*, *Pso04G02820.1* (*PsoCENH3*), *NnCenH3-A*, and *HTR12_ARATH*. EALT is the sequence feature of CENH3 in *P. somniferum* and *P. rhoeas*.

(B) The BlastP alignments between the protein sequences of Prh03G45160.1 and other three CENH3, Pso04G02820.1, NnCenH3-A, HTR12_ARATH. The percents indicate the identities.

Figure R2. The validation of Prh168S1 centromere satellite by PrhCENH3 ChIP-seq.

The Prh168S1, PrhCENH3 ChIP-seq coverage, and the ChIP-seq/input ratio are shown on each chromosome of *P. rhoeas*.

Prh168S1 is the identified centromere satellite in current results
 Prh168S2_H1 is the identified centromere satellite in H1 of the draft haplotype-resolved genome
 Prh168S1_H2 is the identified centromere satellite in H2 of the draft haplotype-resolved genome

Figure R3. Centromere satellites comparison.

- (A) Dotplot show the satellite sequence comparison between Prh168S1 and Prh168S2_H1
 (B) Dotplot show the satellite sequence comparison between Prh168S1 and Prh168S1_H2
 (C) Dotplot show the satellite sequence comparison between Prh168S2_H1 and Prh168S1_H2
 (D) Sequence identity matrix of Prh168S1, Prh168S2_H1, and Prh168S1_H2, obtained by BlastN.

Reviewer #3: *The authors have made detailed revisions to the manuscript in response to the issues. I have no other questions and agree to accept this work.*

RESPONSE: Thanks for your valuable comments for improving our manuscript.